biomechanics

computational fluid dynamics, flow tank torque measurements, hydrodynamics, manoeuvrability, stability, yaw turning

**Author for correspondence:**
Pim G. Boute
e-mail: pim.boute@wur.nl
Tel: +31 (0)50 3632259

# Modulating yaw with an unstable rigid body and a course-stabilizing or steering caudal fin in the yellow boxfish (Ostracion cubicus)

Pim G. Boute[1,2], Sam Van Wassenbergh[3] and Eize J. Stamhuis[1]

[1]Department of Ocean Ecosystems, Energy and Sustainability Research Institute Groningen, Faculty of Science and Engineering, University of Groningen, Nijenborgh 7, 9747 AG Groningen, The Netherlands
[2]Experimental Zoology Group, Department of Animal Sciences, Wageningen University & Research, De Elst 1, 6708 WD Wageningen, The Netherlands
[3]Department of Biology, University of Antwerp, Universiteitsplein 1, 2610 Antwerpen, Belgium

PGB, 0000-0003-2954-635X; SVW, 0000-0001-5746-4621; EJS, 0000-0001-7746-2535

Despite that boxfishes have a rigid carapace that restricts body undulation, they are highly manoeuvrable and manage to swim with remarkably dynamic stability. Recent research has indicated that the rigid body shape of boxfishes shows an inherently unstable response in its rotations caused by course-disturbing flows. Hence, any net stabilizing effect should come from the fishes' fins. The aim of the current study was to determine the effect of the surface area and orientation of the caudal fin on the yaw torque exerted on the yellow boxfish, *Ostracion cubicus*, a square cross-sectional shaped species of boxfish. Yaw torques quantified in a flow tank using a physical model with an attachable closed or open caudal fin at different body and tail angles and at different water flow speeds showed that the caudal fin is crucial for controlling yaw. These flow tank results were confirmed by computational fluid dynamics simulations. The caudal fin acts as both a course-stabilizer and rudder for the naturally unstable rigid body with regard to yaw. Boxfishes seem to use the interaction of the unstable body and active changes in the shape and orientation of the caudal fin to modulate manoeuvrability and stability.

# 1. Introduction

Swimming performance of fishes is determined by motions of the fins and body in combination with the shape of the body and its hydrodynamic properties due to the interplay of thrust and resistance [1,2]. In general, a trade-off is supposed to exist in the morphological adaptations for locomotion depending on the ecological niche of the species. This will result in a body morphology that is specialized to either cruising, accelerating or manoeuvring, which are the three main swimming functions; other fishes are generalists, giving them moderately good performance in all three functions but not superior performance in one [1–6]. According to Webb [1], it is impossible to have all the optimum features of the different swimming types in one fish, which he coins 'the principle of the mutual exclusion of optimum designs'. Consequently, in the course of their evolution, strong selection can be expected on the morphology of the locomotor system of fishes to meet the different swimming function they require given their lifestyle.

Boxfishes (Ostraciidae; Tetraodontiformes), a family of marine fishes, have an exceptional morphology [7–10]. These fishes have evolved a bony encasing of the body, called a carapace [9–11]. This carapace consists of sutured bony hexagonal, pentagonal and heptagonal plates or scutes [10,12] with gaps for the eyes, mouth, gill openings, anus, caudal peduncle with caudal fin, pectoral, anal and dorsal fins. A third up to three-quarters of the length of their bodies is encased in this rigid carapace [13,14]. The shape of the carapace varies widely among ostraciid species, from square to triangular, oblong and pentangular transverse sections to more ellipse-like [15,16]. In addition, the surface of the carapace may be smooth or irregular with protruding spines and is keeled with various ridges [17]. The carapace serves as armour for protection against potential predators [10,18].

The rigidity imposed by the carapace has important consequences for propulsion, stability and manoeuvrability [e.g. 19,20]. Swimming movements can only be derived from their five fins, reaching top speeds of above 6 body lengths s$^{-1}$ using the caudal fin for burst-and-coast swimming [13,21–23]. For slow, rectilinear swimming and manoeuvring, boxfishes use various combinations of the pectoral, dorsal and anal fins for propulsion [19,21], using pectoral/anal-fin-dominant and anal/dorsal-fin-dominant swimming [4,24]. Boxfishes do not migrate and generally move relatively slowly around coral reefs and feed on small benthic organisms [25]. Hence, the abilities of boxfishes to hover, turn in place and even swim upside down are of great importance for the boxfishes' fitness and survival. Despite this unique bauplan, boxfishes are able to execute low recoil motions with near-zero turning radius [11,13,21,23]. This ability to turn and manoeuvre is fully in line with the requirements to move in the complex three-dimensional (3D) coral reef habitat boxfishes inhabit [11,19]. However, the question remains how boxfishes can be highly manoeuvrable without compromising the stability needed for controlled, efficient swimming.

Previous studies hypothesized that the flow over the carapace of boxfishes induces course-stabilizing and self-corrective trimming vortices during pitching (i.e. rotation about the lateral axis) and yawing (i.e. rotation about the dorsoventral axis) [14,15,22,23]. This passive inherent course stabilization by flow over the carapace (without fins) was hypothesized to dampen and counteract perturbations when swimming, and supposed to keep boxfishes on their swimming course while in turbulent waters, resulting in optimal stability [14,22,23,26]. However, the hypothesis concerning overall course stabilization by the carapace was rejected by Van Wassenbergh *et al.* [11]. They showed that the body-induced vortices indeed exist, but that the overall impact on the pitch or yaw torque balance is relatively small: the body (i.e. carapace with a truncated part of the caudal peduncle) of both the yellow boxfish *Ostracion cubicus* Linnaeus, 1758 and smooth trunkfish *Lactophrys triqueter* (Linnaeus, 1758) generates destabilizing hydrodynamic torques (i.e. are naturally hydrodynamically unstable). Consequently, body parts other than the carapace must be responsible for enabling boxfishes to maintain a stable course during swimming.

So how do boxfishes control destabilizing torques on the body to direct their movements? The tight relationship between the angle by which the caudal fin bends to the side and the turning radius indicates that boxfishes, similar to most other fishes, have control over turning by engaging the caudal fin as a rudder [e.g. 6,11,13,19–21,27–37]. It is hypothesized by numerous studies that in a wide range of fish species the caudal fin may be used for steering [6,28–37] and that boxfishes may partially control their turning kinematics by using the caudal fin as a rudder [13,19–21]. Van Wassenbergh *et al.* [11] hypothesized that fins probably play an important role in mitigating this instability and calculated that a caudal fin with a surface area of 0.001 m$^2$ would be able to counteract destabilizing torques of the body itself. Hence, the caudal fin is supposed to be important in turning manoeuvres and thus in controlling instability. In addition, the state of the caudal fin (e.g. completely closed or open) and

angle with respect to the body axis is expected to be important during turning manoeuvres according to qualitative descriptions of living swimming boxfish specimens [13,19] and correlations found between the tail angle and turning radius [20]. Blake [19] observed that during slow forward swimming, the caudal fin remains closed and aligned along the median axis of the fish, while the caudal fin is opened during fast forward swimming. The caudal fin is closed when turning and rotating [19], and the tail rotates in opposite direction compared with the body [20].

The aim of this study was to determine the effect of the caudal peduncle and caudal fin on the unstable yaw properties of the body of *O. cubicus*. To resolve these effects, two physical models of *O. cubicus* were constructed. One model had a reconstructed rigid caudal peduncle while in the other model, the caudal peduncle could be turned and fixed under different angles with respect to the body. In a flow tank, yaw torques were measured at different water flow speeds and with the model at different yaw angles. Firstly, both physical models were tested without caudal fin to quantify the contribution of adding only a caudal peduncle to the carapace. Hereafter, both physical models were provided with an either closed or open reconstructed caudal fin. Additionally, the caudal peduncle and caudal fin were fixed at different angles with respect to the body axis to gain insight in their hydrodynamic interaction. This hydrodynamic interaction is particularly interesting given the wide carapace of boxfishes that may shield the caudal fin from incoming flow, and the above-mentioned vortices that are shed by the carapace under certain angles of attack which may affect yaw torque generated by the caudal fin. Computational fluid dynamics (CFD) simulations were used to verify the data of the physical models and to model the exerted pressures and yaw torque on the body, caudal peduncle and caudal fin separately. We hypothesize that the caudal fin and caudal peduncle are a major means to course-stabilize the hydrodynamically unstable body, but that the effect can be modulated by opening or closing the caudal fin.

# 2. Material and methods

## 2.1. Model species

For this study, the cubical shaped yellow boxfish *O. cubicus* was chosen as model species from the boxfish family (Ostraciidae). This species represents the square cross-sectioned boxfishes. Living as well as dead specimens are relatively easily available, hence this species was used in several previous studies [e.g. 11,38].

## 2.2. Laser scans

The same stereo-lithographic model as used by Van Wassenbergh *et al.* [11] in drag force and yaw torque experiments was used in this study, for more details on 3D scanning, reconstruction and model dimensions we refer to this article [11].

## 2.3. Model, caudal peduncle and caudal fin reconstruction

Two physical models were produced from the laser scan surface using a fused-deposition modelling 3D printing technique (RapMan 3.1 3D Printer Kit, Bits from Bytes, 3D Systems Inc., Rock Hill, USA). Surface irregularities were removed with a plaster-based filler (Alabastine, AkzoNobel) and subsequent sanding. A hole was drilled in the ventral side of the models in which a threaded insert was mounted in such a way that the heart line of the insert was in line with the model's centre of volume. The models were spray-painted with matt black water-resistant paint (Motip EAN 8711347040018). Note that the models, like all boxfish models used in previous research performing force and torque measurements in a flow tank, have a smoother exterior compared with living fish due to the absence of surface scute topography. The effect of adding a natural surface roughness remains untested, but (uniform) modifications of surface roughness constants in CFD models (see below for details on this method) suggest a negligible effect on overall yaw torque [11]. We assumed that the centre of mass was located at the centre of volume, which may introduce some error in the calculation of the relevant yaw torque (i.e. torque about the axis through the centre of mass). However, a previous comparison between experimentally measured centres of mass and the centre of volume based on surface scans of the carapace showed that deviations are limited to a few per cent of carapace length [11].

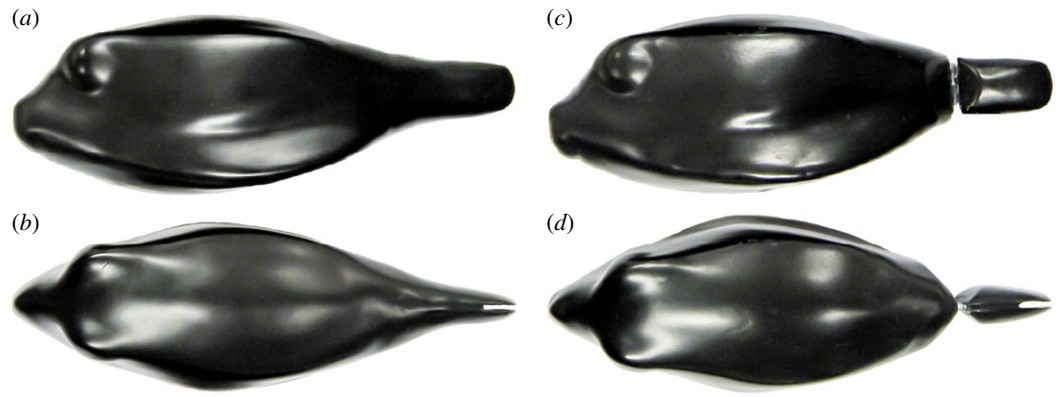

**Figure 1.** Photographs of the physical models of *O. cubicus*. (*a*) Lateral and (*b*) dorsal view of the rigid tail model. (*c*) Lateral and (*d*) dorsal view of the turnable tail model. Note that the caudal peduncle of the turnable tail model has not yet been fixed with knead glue, hence the hinge is visible.

To reconstruct the caudal peduncle and caudal fin for these models, we filmed a representative living *O. cubicus* specimen with a total length of 15.5 cm present in our laboratory seawater aquaria at the University of Groningen, using a camera system at 50 fps (Adimec 1000 m, 1.2 Mpix, Adimec Eindhoven, The Netherlands; Matrox Solios v. 1.0 framegrabber board, Matrox Imaging, Durval, Quebec, Canada) with Realtime Image Recorder software (Ingenica, Lage Zwaluwe, The Netherlands). The caudal peduncle was reconstructed at the physical models with Aquascape Aquarium Epoxy (D-D The Aquarium Solution Ltd) using multiple photographs as reference material. The first model was reconstructed in such a way that the caudal peduncle was fixed with respect to the midline of the model (rigid tail model) (figure 1*a,b*). In the second model, the caudal peduncle was attached to the rest of the carapace using a hinge allowing for it to turn and be fixed at different angles (turnable tail model) (figure 1*c,d*).

To construct the caudal fins, we made photographs of lateral views of the living boxfish with its caudal fin fully open or fully closed. The contours of the caudal fins were outlined using ImageJ 1.42q [39] and isometrically scaled by a few per cent to achieve a matching size of the carapace contours from lateral view. The closed caudal fin and open caudal fin were selected based on having the lowest and highest surface area out of at least five photographs, respectively (figure 2*a,b*). The caudal fin models (closed and open) were cut from 1 mm thick poly(methyl 2-methylpropenoate) (PMMA) sheet with Young's modulus of approximately 3 GPa and sanded at the edges (figure 2*c,d*). Since this material is relatively stiff but still bends a bit at larger forces, it was considered to be a reasonable mimic of a caudal fin, though commonly observed types of morphing of the fins due to the interaction with the water, fin-ray geometry and the fin's material properties, such as cupping [40], could not be included in our simplified model. Bending of the caudal fin was not quantified, but bending was minimal (estimated less than 3 mm deflection of the caudal fin tip) and only occurred at the highest flow speed when the caudal fin was oriented relatively perpendicularly to the incoming flow. A vertical saw cut was made in the middle of the caudal peduncles of the physical models of both fish allowing the caudal fins to be attached to the models in an exchangeable way using knead glue (Poster Buddies, Pritt, Henkel AG & Co., Germany). The caudal peduncle of the turnable tail model could be fixed at any angle of attack (angle $\theta$; figure 3) with respect to the model's midline using knead glue to fill up the space around the hinge and a copper wire reinforcement to prevent deformation during the experiment. The frontal surface areas were 21.7 and 21.1 cm$^2$ for the rigid tail model and turnable tail model, respectively. Both models had a volume of 0.16 dm$^3$ and were 16.95 cm long with caudal fin. The lateral view surface area of the closed caudal fin was 8.23 cm$^2$ and for the open caudal fin 19.83 cm$^2$, respectively.

The 3D laser scan CFD model as used in Van Wassenbergh *et al.* [11] was modified by adding a caudal peduncle and caudal fin in accordance with aforementioned procedures. The caudal peduncle and caudal fin of the physical models were mimicked using external contour landmark digitizations based on photographed fish. The triangulated surface was converted into a smooth Non-Uniform Rational B-Spline surface (NURBS, VRMesh 5.0, VirtualGrid, Bellevue, WA, USA) (figure 4), enabling calculation of the hydrodynamic pressures and torques exerted separately on the body, caudal peduncle and caudal fin. In all CFD simulations, the space between the body and caudal peduncle was not filled up as was done in the physical turnable tail model.

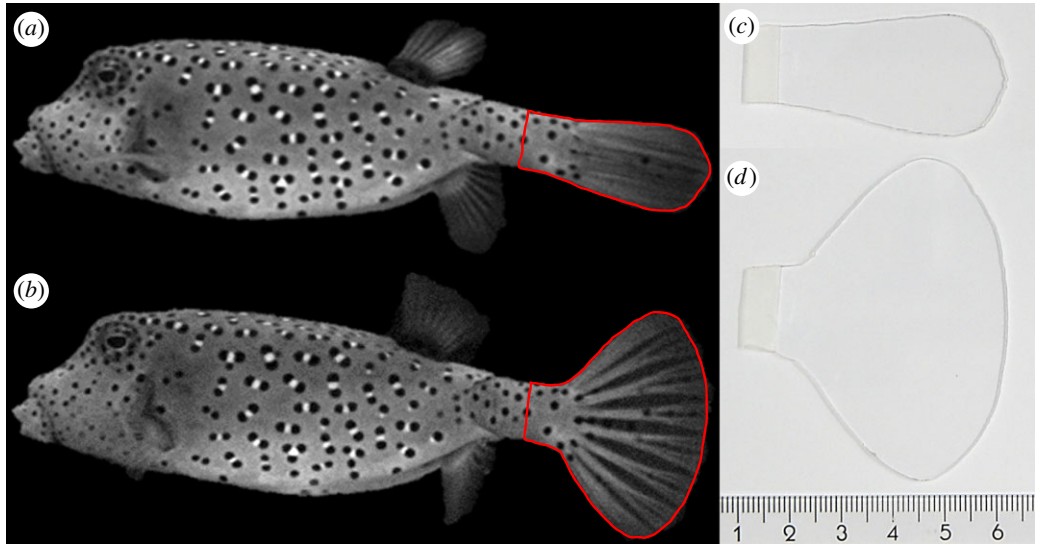

**Figure 2.** Photographs of the living *O. cubicus* specimen in lateral view with (*a*) closed caudal fin and (*b*) open caudal fin from which the contours were extracted (red line). These contours were isometrically scaled to construct the (*c*) closed caudal fin (8.226 cm$^2$) and (*d*) open caudal fin (19.828 cm$^2$) which were attached to the physical models of *O. cubicus*. Scale bar is in centimetres and only applicable to the right panels.

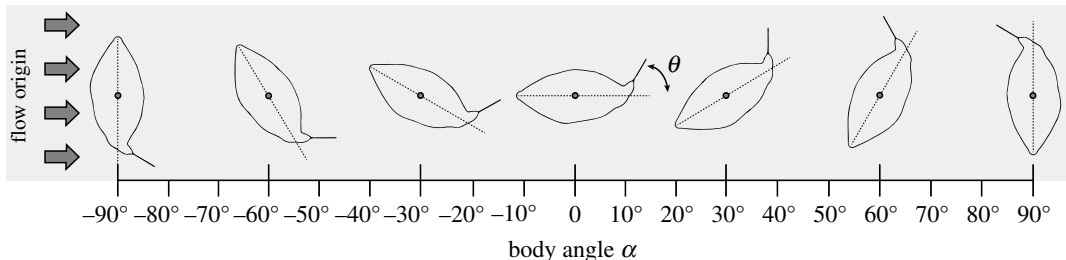

**Figure 3.** Schematic of the dorsal view of the experimental set-up. The outline of the *O. cubicus* models is shown. The arrows indicate the direction of the flow, coming from the left. The small circle shows the position of the centre of volume and point where the metal rod was attached to the model, hence the point at which the complete model could be rotated (fulcrum). The models were fixed at body angles $\alpha$ varying from −90° up to 90° with 10° increments. The −90° to 0° range and 0° to 90° range were measured in separate series. The rigid tail model had the tail permanently fixed at an angle $\theta$ of 0° relative to the midline axis of the body (dotted line). The turnable tail model had the tail fixed at the angles $\theta$ of 0°, 10°, 20°, 30° and 40°. Note that positive tail angles $\theta$ correspond to deflection in the same, counterclockwise direction as positive body angles $\alpha$.

## 2.4. Flow tank measurements

The yaw torque measurements were performed in a 300 l flow tank containing fresh water at 20°C as described in Van Wassenbergh *et al.* [11]. The dimensions of the rectangular test section with laminar flow were 25 × 25 cm in cross section and 50 cm in length. A transparent PMMA plate covered the top of the test section to prevent wave formation. During the measurements, we mounted the models on a 5 mm diameter stainless steel rod which was attached to a fixed measuring platform above the tank, as used by Van Wassenbergh *et al.* [11], so that the centre of volume of the model was exactly in the middle of the cross section of the tank.

The rod was mounted in a custom-made ball bearing seat that served as an axle that could rotate freely and was practically frictionless. After setting the angle of the model relative to the flow (body angle $\alpha$; figure 3), the torque could be measured as a function of flow speed using a 40 mm arm that was fixed to the axle, using a force sensor (Vernier Dual range 10 N/50 N, DFS-BTA, Vernier Software & Technology, Beaverton, USA) which was connected to a PC using a Vernier LabPro Interface. Data were recorded using Vernier LoggerPro 3.8 software at a sampling frequency of 50 Hz after calibration.

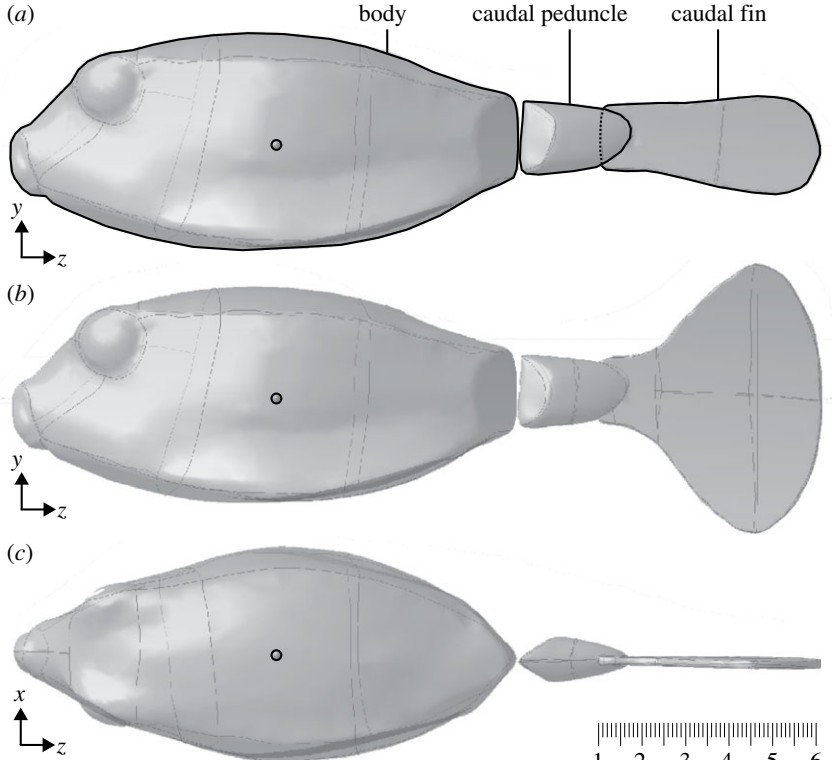

**Figure 4.** Geometry of the *O. cubicus* models used in computational fluid dynamics (CFD) simulations. (*a*) Lateral view of the model with closed caudal fin. (*b*) Lateral view of the model with open caudal fin. (*c*) Dorsal view of the model. The small circle shows the position of the centre of volume. The subdivision of the boxfish's surface in body, caudal peduncle and caudal fin is indicated at the top. Scale bar is in centimetres.

In the first set of experiments, no caudal fins were attached to the *O. cubicus* models and the tail of the turnable tail model was fixed at an angle $\theta$ of 0° with respect to the midline axis of the model's body. The models were positioned in the flow tank at different body yaw angles $\alpha$, namely, 0°–90° with 10° increments (figure 3). The flow speed was varied between 0 and 0.5 m s$^{-1}$ (approx. 3.5 body lengths s$^{-1}$) with 0.1 m s$^{-1}$ increments. The maximum flow speed was estimated to be representative of fast swimming in relatively large individuals [22]. These speeds also correspond roughly to flow speeds measured in coral reef habitats [41–44] which overlap with the yellow boxfish distribution according to Myers [25]. Five repeats per speed per body angle $\alpha$ were recorded. A single recording lasted 10 s of continuous recording and consisted of 501 data points. Finally, the five replicates were averaged and standard deviations were calculated.

Subsequently, the experiments were repeated with both *O. cubicus* models with an artificial closed caudal fin and open caudal fin attached, fixed at an angle $\theta$ of 0° (figure 3). Measurements were recorded at body angles $\alpha$ for 0° up to and including 60° with 10° increments. Reliable measurements at angles above 60° were not possible because of wall effects since the caudal fin was too close to the flow tank wall; the minimum distance was kept at 15 mm which was confirmed by particle image velocimetry to have no wall effects.

Finally, the experiments were repeated with the turnable tail model with the tail fixed at different angles $\theta$ (figure 3), respectively 10°, 20°, 30° and 40° with closed and open caudal fins, applying the same measurement procedure as described above. The −90°–0° and 0°–90° ranges were measured in separate series for tail angles $\theta$ of 10°, 20°, 30° and 40°, resulting in twice the number of data points for body angles $\alpha$ of 0°.

For the experiments of the rigid tail model and turnable tail model with tail angle $\theta$ of 0°, a yaw torque acting to increase the angle between flow and the midline of the model was identified as positive. Yaw torque is defined as positive when they act in the counterclockwise direction from a dorsal view on the boxfish (figure 3). For $\alpha > 0$, positive torques are course destabilizing, i.e. increasing the angle between the flow and the midline of the model. For $\alpha < 0$, positive torques are course stabilizing, i.e. decreasing the angle between the flow and the model's midline.

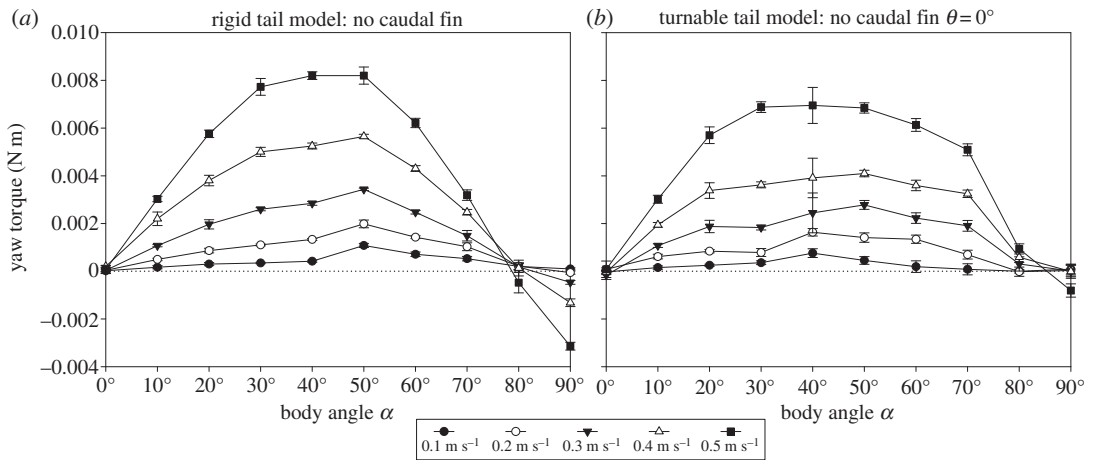

**Figure 5.** Performance of the (*a*) rigid tail model and (*b*) turnable tail model (tail angle $\theta$ of 0°) without caudal fin. Yaw torque (N m) about the centre of volume per body angle $\alpha$ (°) for *O. cubicus*. Experimental data are given for five flow speeds between 0.1 m s$^{-1}$ and 0.5 m s$^{-1}$ with 0.1 m s$^{-1}$ increments. The dotted lines indicate a net yaw torque of 0 N m. Error bars shown for all flow speeds denote the between-repeat standard deviation.

## 2.5. Computational fluid dynamics (CFD) simulations

The CFD surface model was imported into ANSYS DesignModeler 14.5.7 (ANSYS Inc., Canonsburg, PA, USA) and placed inside a cylindrical outer boundary of the flow domain. The angle of the body and tail were specified as described above. The flow domain was meshed in ANSYS Meshing 14.5.7 and imported into ANSYS Fluent 14.5.7 in which the boundary conditions were set. The size of the mesh elements, position of the boxfish model in the flow domain, dimensions of the outer boundary of the flow domain, boundary conditions and solver settings were the same as in Van Wassenbergh *et al.* [11]. All torques resulting from the CFD analysis are based on both shear and pressure forces. CFD allows for the subdivision of forces and torques of body, caudal peduncle and caudal fin to evaluate their relative contribution to the total torque. In the first set of CFD measurements, the model was placed at body angles $\alpha$ of −60°, −40°, −20°, 0°, 20°, 40° and 60° with a tail angle $\theta$ of 20° with the closed caudal fin and open caudal fin at a flow speed of 0.5 m s$^{-1}$ to verify the flow tank experimental data (figure 3). Secondly, the CFD model was fixed at a body angle $\alpha$ of 20° with varying tail angles $\theta$ of −60°, −40°, −20°, 0°, 20°, 40° and 60° (figure 3).

# 3. Results

## 3.1. Rigid tail model and turnable tail model: no caudal fin at tail angle $\theta$ of 0°

The yaw torques about the centre of volume in relation to body angle showed a similar pattern for both the rigid tail model and turnable tail model when no caudal fin was attached (figure 5). When the body angle is increased from its original orientation in line with the flow (body angle $\alpha$ of 0°), the yaw torque about the centre of volume of the body becomes increasingly positive (i.e. rotational direction away from the flow or destabilizing) to reach a peak for the rigid tail model at either 40° and 50° (flow tank at 0.5 m s$^{-1}$) or 50° (flow tank up to 0.4 m s$^{-1}$) (figure 5*a*), and for the turnable tail model at either 30° and 40° (flow tank at 0.5 m s$^{-1}$) or 50° (flow tank up to 0.4 m s$^{-1}$) (figure 5*b*). For larger body angles, the yaw torques decrease but remain destabilizing until a body angle of 80° (figure 5). For the rigid tail model and to a lesser extent for the turnable tail model, the torque becomes negative (i.e. rotational direction towards the flow or stabilizing) at a body angle of 90°. Larger flow speeds generate larger yaw torques.

## 3.2. Rigid tail model and turnable tail model: closed caudal fin and open caudal fin at tail angle $\theta$ of 0°

Yaw torques about the centre of volume in relation to body angle showed a similar pattern for the rigid tail model and turnable tail model with the closed caudal fin and open caudal fin at tail angle $\theta$ of 0°

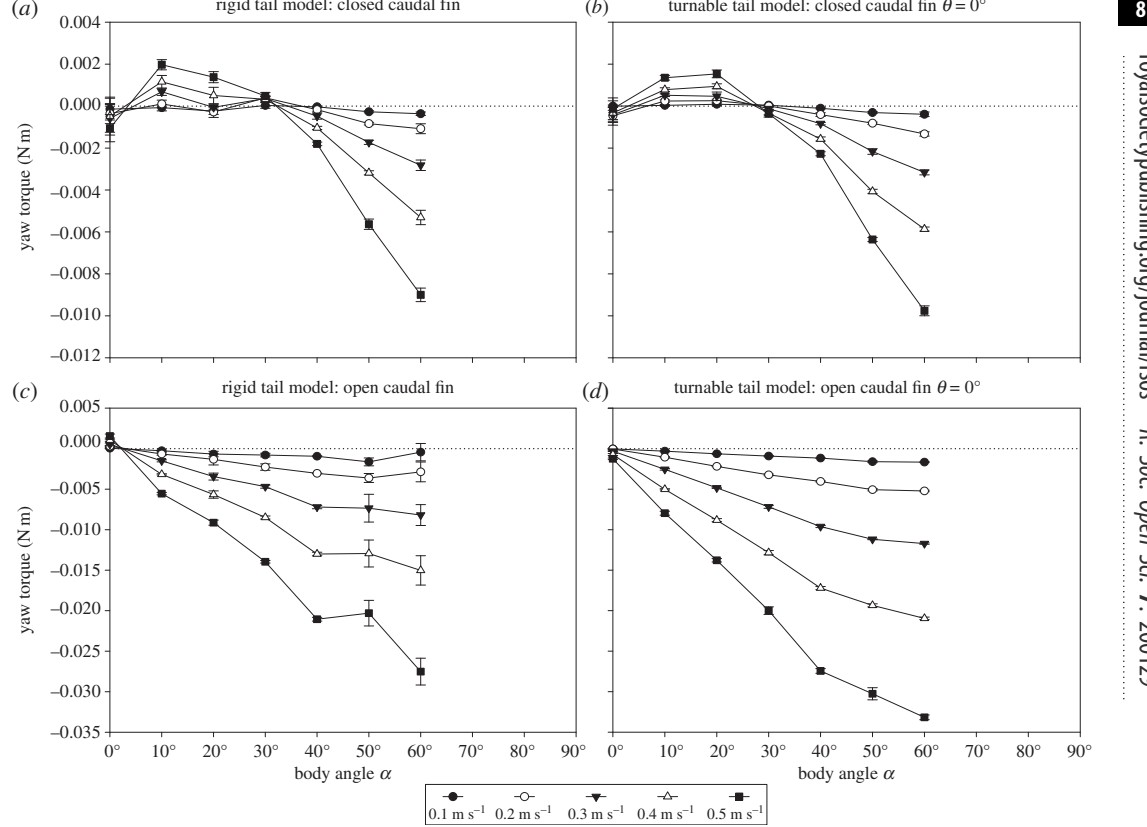

**Figure 6.** Performance of the (*a,c*) rigid tail model and (*b,d*) turnable tail model with a (*a,b*) closed caudal fin and (*c,d*) open caudal fin at a tail angle $\theta$ of 0°. Yaw torque (N m) about the centre of volume per body angle $\alpha$ (°) for *O. cubicus*. Experimental data are given for five flow speeds between 0.1 and 0.5 m s$^{-1}$ with 0.1 m s$^{-1}$ increments. The dotted lines indicate a net yaw torque of 0 N m. Error bars shown for all flow speeds denote the between-repeat standard deviation.

(figure 6). When the body angle is increased from its original orientation in line with the flow (body angle $\alpha$ of 0°) of both models with a closed caudal fin, the yaw torque about the centre of volume of the body becomes increasingly positive (i.e. rotational direction away from the flow or destabilizing) to reach a peak for the rigid tail model at either 10° (flow tank at all speeds) (figure 6*a*) and at 20° for the turnable tail model (flow tank at all speeds) (figure 6*b*). At larger body angles, the yaw torque decreases to 0 N m at a body angle around 30° for both models and becomes increasingly negative (i.e. rotational direction towards the flow or stabilizing) for both models (all speeds) (figure 6*a,b*). When the body angle is increased from its original orientation in line with the flow (body angle $\alpha$ of 0°) of both models with an open caudal fin, the yaw torque about the centre of volume of the body becomes increasingly negative (i.e. rotational direction towards the flow or stabilizing) (figure 6*c,d*).

## 3.3. Turnable tail model: closed caudal fin and open caudal fin at tail angles $\theta$ of 0°, 10°, 20°, 30° and 40°

The steering effect of the closed caudal fin and open caudal fin on the *O. cubicus* model at different body and tail angles is evident in figure 7. The *y*-axes are the same scale in all graphs for easier comparison. Yaw torques about the centre of volume in relation to the body angle showed a similar pattern for the turnable tail model with tail angles $\theta$ of 0°, 10°, 20°, 30° and 40° for the closed caudal fin (figure 7*a,c,e,g* and *i*) and open caudal fin (figure 7*b,d,f,h* and *j*). The yaw torque in relation to tail angle $\theta$ becomes larger with increasing flow speed for both the closed caudal fin and open caudal fin at all body angles $\alpha$. An open caudal fin generates larger yawing torques than a closed caudal fin. The data for the yaw torques of the turnable tail model with the tail angle $\theta$ of 0° are the same as in figure 6*b,d*. However, for comparison purposes, the data were mirrored over the *y*-axis and *x*-axis of 0 (light grey area).

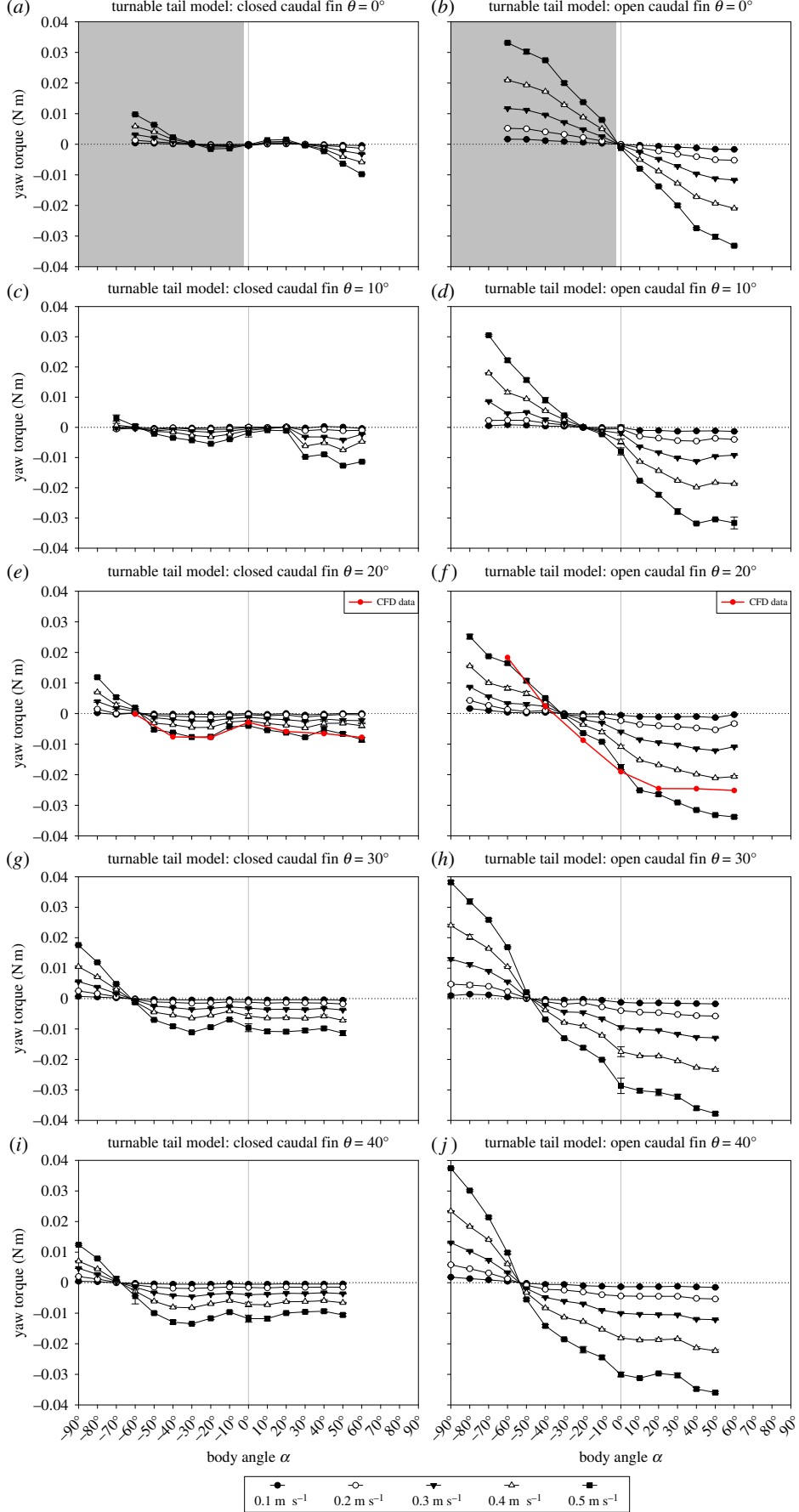

**Figure 7.** (*Caption overleaf.*)

**Figure 7.** (*Overleaf.*) Performance of the turnable tail model with a (*a,c,e,g* and *i*) closed caudal fin and (*b,d,f,h* and *j*) open caudal fin at the tail angles $\theta$ of 0°, 10°, 20°, 30° and 40°. Yaw torque (N m) about the centre of volume per body angle $\alpha$ (°) for *O. cubicus*. Experimental data are given for five flow speeds between 0.1 and 0.5 m s$^{-1}$ with 0.1 m s$^{-1}$ increments. (*a*) and (*b*) are based on data figure 6 (*b,d*). The left-hand grey areas in (*a*) and (*b*) indicate mirrored data over the *y*-axis and *x*-axis from the right-hand white area. The *y*-axes are the same scale in all graphs for easier comparison. The vertical grey lines indicate a body angle $\alpha$ of 0°. The red lines in (*e*) and (*f*) show the CFD simulation data for the body angles $\alpha$ of −60°, −40°, −20°, 0°, 20°, 40° and 60° at a tail angle $\theta$ of 20° with a (*e*) closed or (*f*) open caudal fin at 0.5 m s$^{-1}$. The dotted lines indicate a net yaw torque of 0 N m. Error bars shown for all flow speeds denote the between-repeat standard deviation.

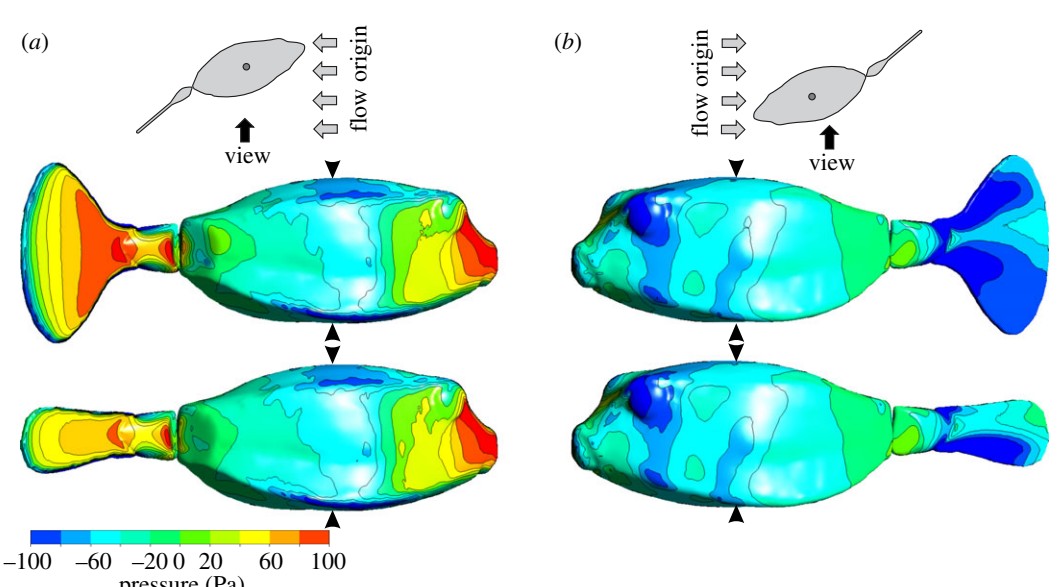

**Figure 8.** CFD simulations showing exerted pressure on (*a*) the flow-facing side and (*b*) the non-flow facing side of *O. cubicus* with open caudal fin (middle panels) and closed caudal fin (bottom panels) at a flow speed of 0.5 m s$^{-1}$. The body angle $\alpha$ and tail angle $\theta$ are both 20°. The small circle in the grey line drawings at the top shows the position of the centre of volume in dorsal view. The black arrowheads show the position of the axis of rotation, i.e. going through the centre of volume.

When the caudal fin is closed, the switch point between right and left turning (i.e. the point where the yaw torque is 0 N m, i.e. neutral; determined by linear interpolation between the sample points with positive and negative torque) shifts towards more negative body angles when the tail angle is increased for all flow speeds (i.e. from a set body angle $\alpha$ of −27° at a tail angle $\theta$ of 0°, $\alpha$ = −57° at $\theta$ = 10°, $\alpha$ = −57° at $\theta$ = 20°, $\alpha$ = −62° at $\theta$ = 30°, to $\alpha$ = −68° at $\theta$ = 40°; figure 7*a,c,e,g* and *i*). The yaw torques in relation to body angle become larger with increasing flow speed (maximum values of −0.014 and 0.018 N m at a flow speed of 0.5 m s$^{-1}$), although yaw torques do not necessarily increase with increasing body angles. When the caudal fin is opened, the switch point between right and left turning (i.e. the point where the yaw torque is 0 N m, i.e. neutral) shifts towards more negative body angles when the tail angle is increased for all flow speeds (i.e. from a set body angle $\alpha$ of 0° at tail angle $\theta$ = 0°, $\alpha$ = −20° at $\theta$ = 10°, $\alpha$ = −32° at $\theta$ = 20°, $\alpha$ = −48° at $\theta$ = 30°, to $\alpha$ = −54° at $\theta$ = 40°; figure 7*b, d,f,h* and *j*). The yaw torques in relation to body angle become larger with increasing flow speed (maximum values of −0.038 and 0.038 N m at a flow speed of 0.5 m s$^{-1}$), although yaw torques do not necessarily increase with increasing body angles. The CFD simulation yaw torque data fit the experimental results very well (figure 7*e,f*: red lines).

## 3.4. Hydrodynamic forces on the body, caudal peduncle and caudal fin

The pressure distribution on the yellow boxfish model at a body angle $\alpha$ and tail angle $\theta$ of 20° shows the difference between the closed caudal fin and open caudal fin (figure 8). When the caudal fin is closed, the area of high positive pressure on the flow-facing side of the caudal fin is significantly smaller than when the caudal fin is open, irrespective of tail angle $\theta$ (figures 8 and 9). Therefore, the torque due to

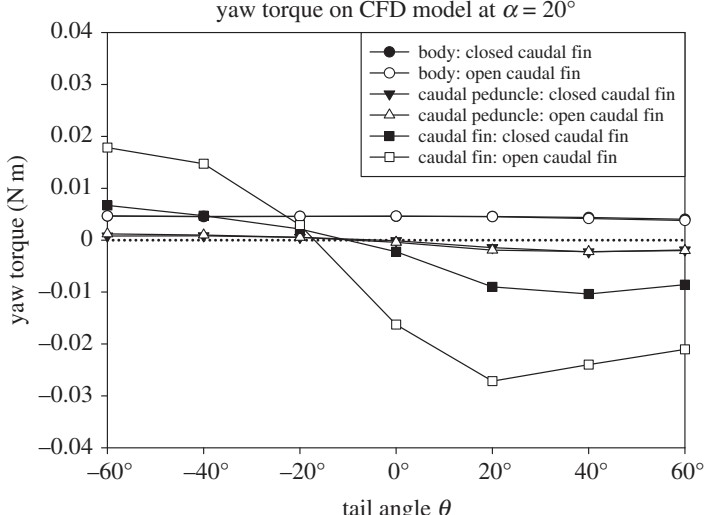

**Figure 9.** CFD simulation data showing the individual yaw torques on body, caudal peduncle and caudal fin of the *O. cubicus* model with a closed or open caudal fin at a flow speed of 0.5 m s$^{-1}$. The body angle $\alpha$ is 20° and the tail angles $\theta$ are −60°, −40°, −20°, 0°, 20°, 40° and 60°. Both lines of the body and caudal peduncle overlap. The dotted line indicates a net yaw torque of 0 N m.

the caudal fin is larger when opened compared with closed. Depending on the total torque on the model, resulting from the exerted pressure distribution (figure 9), the model will turn either left or right. In the example shown in figure 8*a*, the net torque is −0.006 N m (figure 7*e*), i.e. almost neutral but just right turning, while in figure 8*b*, the net torque is −0.026 N m (figure 7*f*), i.e. right turning towards the oncoming flow. The pressure distribution on the body does not depend on the state of the caudal fin (closed or open). Consequently, the net torque on the fish model at a given body and tail angle depends on the torque by the caudal fin, which in turn depends on its surface area (figures 8 and 9). The relative effect of the caudal peduncle on torque is minimal (figures 8 and 9) at $\alpha = 20°$ and $\theta = 20°$, for example, its stabilizing torque is only 20% and 7% of that of the closed caudal fin and open caudal fin, respectively.

## 4. Discussion and conclusion

*Ostracion cubicus* inhabits the waters of tropical coral reefs and manoeuvres in crevices and around reef, rock and sand patches feeding on a variety of benthic organisms [24]. Hence, it has been hypothesized that its morphological traits enhance manoeuvrability [11]. Van Wassenbergh *et al.* [11] showed that the hydrodynamic torques on the body worked in a destabilizing manner for both the triangular cross-sectioned *L. triqueter* and for the square cross-sectioned *O. cubicus*. The models used by Van Wassenbergh *et al.* [11] lacked fins and included only a small part of the caudal peduncle. They suggested that, in analogy with other fishes [e.g. 6,28–37], the caudal fin could play an important role in controlling yaw in boxfishes.

The results of the current study show that the caudal fin is indeed a major means to course-stabilize the hydrodynamically unstable body of *O. cubicus* (figures 5 and 6). The caudal fin has been hypothesized many times to play a central role in controlling turning manoeuvres in many fish species as well as in boxfishes [e.g. 6,13,19–21,28–37]. This course stabilization was shown by measuring yaw torque at different body angles ($\alpha$) in a flow tank with a physical model of *O. cubicus* with an attached caudal peduncle and caudal fin. The results corroborate measurements and computations by Van Wassenbergh *et al.* [11] (figure 5*c*). Both our rigid tail model and turnable tail model are unstable without caudal fin, which corresponds to the model used by Van Wassenbergh *et al.* [11] and show larger destabilizing yaw torques at higher flow speeds (figure 5). Moreover, generated yaw torques of both models were highly similar (figure 5). Since the caudal peduncle was present in both the rigid tail model and turnable tail model when no caudal fin was attached, we conclude that the caudal peduncle on its own has no major impact on the course stabilization of *O. cubicus*, which is confirmed by the CFD measurements (figure 9).

Our results illustrate an interplay between a destabilizing body and a course-stabilizing caudal fin. When a closed caudal fin is attached to any of two models at tail angle $\theta$ of 0°, unstable yaw torques are likely to occur at body angles of 10° and 20° as a result of this interplay (figure 6$a$,$b$). The body has a greater impact at low body angles than the closed caudal fin, the latter becomes more dominant when the body angle increases. When the caudal fin is open, yaw torques are course-stabilizing at all body yaw angles (figures 6$c$,$d$) indicating that the open caudal fin is dominant over the net destabilizing body. Consequently, opening or closing the caudal fin significantly changes the nature of the torque balance due to water flow over the boxfish.

The same trend is discernible from the experiments in which the caudal peduncle and caudal fin were fixed at different tail angles (turnable tail model) and at different body angles with both a closed caudal fin and open caudal fin (figure 7). These results show that *O. cubicus* is able to turn by using the interplay of the destabilizing body and the course-stabilizing caudal fin by opening or closing the caudal fin and turning the tail. When the caudal fin is closed, the body has a more dominant effect on the overall yaw torque. When the caudal fin is open, the destabilizing effect of the body is strongly counteracted. For illustration, when the tail angle $\theta$ is 20°, the switch point of the model with an open caudal fin is around a body angle $\alpha$ of −32° (figure 7$f$). This is the net effect of the sum of the body and caudal fin. This switch point would theoretically be expected at a body angle $\alpha$ of −20° when the tail angle $\theta$ is 20°, if the body would be hydrodynamically neutral. However, the body induces destabilizing torques (i.e. −0.0056 N m at a body angle $\alpha$ of −20° with a flow speed of 0.5 m s$^{-1}$ (figure 5$b$); note that the values for negative body angles $\alpha$ are not shown but are expected to follow the same pattern as the positive body angles but mirrored over the $x$-axis). This torque is about the same at a body angle $\alpha$ of −20° when the tail angle $\theta$ is 20° with an open caudal fin (figure 7$f$). When the caudal fin is closed, the net unstable body is more dominant at smaller body angles since the caudal fin is in the wake of the body. At large body angles, the tail adds extra yaw since then the caudal fin extends from behind the body into the undisturbed water flow (figure 9). This results in a switch point at a body angle $\alpha$ of around −57° when the tail angle $\theta$ is 20° (figure 7$e$). It should be noted, however, that the exact angle values of the switch points may depend on the position of the centre of mass, which could only be estimated and not accurately measured.

These results are evidence for a major effect of the caudal fin on course-stabilizing the hydrodynamically unstable body of *O. cubicus*. In addition, CFD simulations gave more insight in the pressure distribution and net yaw torque on the body, caudal peduncle and caudal fin (closed and open) separately. These simulations confirm our experimental findings. During large positive body angles $\alpha$, the static equilibrium of hydrodynamic forces about the centre of volume on the boxfish's body, caudal peduncle and caudal fin, is dominated by pressure-induced forces on the caudal fin (figures 7$e$,$f$ and 9). The exerted pressure on the caudal fin is net overruling the positive pressure on the head (figure 8). This induces the observed right steering at large positive body angles $\alpha$. When the model is at the switch point, pressures on the caudal fin and caudal peduncle compared with the head are equal, resulting in a net yaw torque of 0 N m (i.e. neutral). When the model is positioned at large negative body angles $\alpha$, pressure on the caudal fin is larger than on the head, resulting in leftward steering. CFD results support the experimental findings that the caudal fin is a major contributor to steering for *O. cubicus* depending on caudal fin state and tail angle $\theta$ (figure 9).

Our results indicate that boxfishes are (actively) controlling instability, rather than relying on hydrodynamic stability inherent to rigid structures of their body which they would need to correct for when starting a manoeuvre. The original idea that the carapace of boxfishes may act self-stabilizing in the flow (i.e. automatic re-aligning the body to the swimming direction) [14,15,22,23] would imply the need for increased forces exerted by the fins to turn the fish while swimming to cancel out the self-stabilizing torque on the carapace. However, the entire carapace acts destabilizing for yaw (figure 5) and pitch [11], as reviewed in the News and Views article of Farina & Summers [45]. Hence, performing turning manoeuvres will require less force from the fins or will result in faster turns for a given force input from the fins. The downside of the latter situation is that recoil motions are more prone to occur during swimming, and these can be energetically costly [2]. The previously discovered vortices that cause self-corrective forces on the carapace [14,15,22,23] will contribute to reduce such recoil motions. Nevertheless, a trade-off exists between energetically costly recoil motions during routine swimming (when controlling instability) versus work required to perform manoeuvres (when correcting for stability), and the selected compromise will depend on the fitness consequences for the species [2]. The reason why boxfishes evolved a locomotor system that controls instability probably relates to their manoeuvrability-demanding lifestyle.

In general, animals may use diverse means to avoid predation [46]. Increased manoeuvrability may decrease potential predation risks, hence increase survival [2,5,6,31,33,47–51]. The anti-predator defence in form of a rigid, non-streamlined [10,32] carapace comes at a cost of increased drag force when swimming or station holding [11,13]. Additionally, higher drag may come at the cost of higher predation risk [51–53]. However, many boxfish species also employ a chemical defence secreted in mucus when stressed [54–56]. The chemical, a potent toxin known as ostracitoxin, is an ichthyotoxic skin secretion cocktail [57–59]. The chemical defence system of *O. cubicus* in combination with the non-flexible and relatively broad carapace restricting refuge seeking in the coral reefs' crevices may serve as non-locomotor, anti-predator defence mechanism indicating that *O. cubicus* reduces potential refuging behaviour costs [26,60–62]. The bright coloration of *O. cubicus*, which is more pronounced in juveniles, might indicate an early development of the chemical defence mechanism [63], i.e. a form of aposematism [64]. Juvenile and adult yellow boxfish differ to some extent in body shape; it remains unclear if the relative role of body and caudal fin in manoeuvring is similar in juvenile and adult boxfish despite the different body shape or proportions or both.

We hypothesize that the caudal fin also controls yaw in other Ostraciidae species, the closely related deepwater boxfishes (Aracanidae) and probably also in other relatively rigid-bodied tetraodontiform fishes such as smooth pufferfishes (Tetraodontidae) [33,35] and porcupinefishes (Diodontidae) [37,65] exhibiting median-paired fin locomotion [12,16,17,66,67] to increase manoeuvrability. This has also indicatively been shown in previous studies [29–21,28,33,35,37]. Blake [19] qualitatively linked posture and state of the caudal fin to yaw, roll and pitch movements for the longhorn cowfish *Lactoria cornuta* (Linnaeus, 1758) and the humpback turretfish *Tetrosomus gibbosus* (Linnaeus, 1758). During slow forward swimming, the caudal fin remains in a closed position and is aligned along the median axis while making small lateral excursions that counter lateral deflections of the body [19]. During fast forward swimming, the caudal fin is opened and moves laterally from the median plane in an arc of about 35° [19]. The state of the caudal fin was, therefore, hypothesized to contribute to swimming course stability [11], which is in line with our quantitative findings in static models of *O. cubicus*. The seemingly active lateral movements of the caudal fin during fast swimming as observed by Gordon [13] and Blake [19] indicates that body/caudal-fin propulsion is used, which was originally termed by Breder [28] as ostraciiform locomotion [68]. However, apart from fast burst-and-coast swimming, propulsion is median/paired-fin driven, engaging the caudal fin as rudder [13,19,20,28]. This role of the caudal fin in swimming is in accordance with the large yaw torques generated by the caudal fin as measured in the current study. Walker [20] studied movement of the tail in relation to the body of the white-spotted boxfish *Ostracion meleagris* G. Shaw, 1796 and found that the turning kinematics are partially controlled by using the caudal fin as a rudder. During turning and rotation, the caudal fin is closed [19] and the tail rotates in opposite direction compared with the body [20], matching our findings. Although the caudal fin is a major means for *O. cubicus* to control the hydrodynamically unstable body, one should consider that the paired and median fins may also play an important role in controlling body yaw, as well as pitch and roll [5,13,19,21,28–30,33,35–37,50,65,69–71].

In conclusion, the results of this study quantitatively show that actively changing the shape and orientation of the caudal fin plays an important role in controlling yaw torque in yellow boxfish (*O. cubicus*). The caudal fin is both a course-stabilizer and a rudder. These findings match with the yellow boxfish's swimming behaviour in its natural habitat. Although this new information is an important first step in improving our understanding in the control of yaw movements in boxfishes, further study is needed to unravel how all components of the boxfishes' locomotor apparatus function together, from a dynamic perspective, during lateral gust flows and turning.

Ethics. No animal experiments were conducted during this study since filming the living yellow boxfish (*Ostracion cubicus*) specimen was not considered an animal experiment.

Data accessibility. All empirically measured torque data are given in the figures of the article. The IGES and CFD data files are deposited at the Dryad Digital Repository: https://doi.org/10.5061/dryad.dz08kprsv [72].

Authors' contributions. E.J.S. and S.V.W. conceived and designed the study. P.G.B. constructed the physical models of the yellow boxfish and carried out the flow tank measurements. S.V.W. performed and analysed the CFD simulations. P.G.B. and E.J.S. analysed and interpreted the results. All authors contributed to writing the manuscript and gave final approval for publication.

Competing interests. We declare that we have no competing interests.

Funding. This research was supported by a grant from the University of Antwerp (grant no. BOF/KP 24346) to S.V.W.

Acknowledgements. We thank both Dennis Worst and Klaas van Manen for their technical assistance during this study. We also thank Tina A. Marcroft and Michael E. Alfaro for allowing us to use their 3D laser scan for this study.

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
