## [Reviewer comments · Royal Society Open Science]

Review History

RSOS-191335.R0 (Original submission)

Review form: Reviewer 1

Is the manuscript scientifically sound in its present form?

Yes

Are the interpretations and conclusions justified by the results?

No

Is the language acceptable?

Yes

Do you have any ethical concerns with this paper?

Yes

Have you any concerns about statistical analyses in this paper?

No

Recommendation?

Accept with minor revision (please list in comments)

Comments to the Author(s)

This article about the hydrodynamics of the caudal fin of the yellow boxfish is an interesting study that offers some insights into the utility of the large caudal fin of boxfishes, in coordination with the rigid carapace. Using models, the authors make a compelling case that the caudal fin could act as a stabilizer in response to incoming flow.

The revisions made by the authors based on feedback from previous reviewers have substantially improved this manuscript. My concerns are primarily about the framing of this paper, both in terms of broadening its appeal for this journal and in terms of adequately putting the results into context.

Major comments:

I feel that the biggest weakness of this study is that there have been so few studies on the use of the caudal fin in living boxfishes. Detailed descriptions of when and how boxfishes use their caudal fins are necessary to interpret the modeling results of this study. This is a principle that applies to all modeling studies - while models are very informative, they are best interpreted in the context of how the animal is using the structure in life. For example, the authors show clear differences in yaw torque between an open and closed caudal fins, but there is no documentation of when boxfishes use an open or closed caudal fin. The only live boxfish studies that have included analysis of caudal fin motions that I am aware of are Walker (2000) and Gordon et al. (2000), which the authors cite, but these studies did not document whether the fin was open or closed during turning or burst swimming.

I understand that it is likely outside the scope of this study for the authors to address this, but I feel that a lack of live animal observations is a limitation of this study that reduces its potential impact. The authors state that they have these animals in the lab, so it seems that a reasonable first step would have been to simply document how the tail is being used (open vs. closed caudal fin, tail position during steering vs. stabilizing, etc.) before proceeding to models. From watching videos from aquarists online, I could easily imagine that boxfishes are using their caudal fin as a rudder, exactly as the authors suggest (they are certainly not using it in the same way as a bluegill). However, without proper documentation of these behaviors, much of the interpretation in the discussion is speculation. Simple experiments could be done to destabilize a living boxfish with bursts of water and observe how the caudal fin is used to respond to this. Even video analysis of a free-swimming animal in a complicated flow environment could be informative.

I also feel that the authors limit themselves by using primarily a "stability vs. maneuverability" framework in their writing. There is a much bigger insight to be gained here, which is that we have been misinterpreting "ostraciiform" locomotion for years. Walker (2000) and Gordon et al. (2000) made this point by clearly demonstrating that the caudal fin is rarely used for propulsion except in short burst-and-coast behaviors, which is in contrast with Breder's classification of ostraciiform locomotion as being caudal-driven. Walker, Gordon, and others have set up an excellent framework for this study: there is a continuum among fishes of caudal fins being used as a primarily as a propulsor or primarily as a rudder, and boxfishes represent an extreme on this spectrum. So, how does the use of the caudal fin in boxfishes differ from that of other fishes? How are the fluid dynamics of steering fundamentally different from propulsion, and how are the boxfishes taking advantage of this? This is an issue of framing. The authors give a very slight nod to these ideas (lines 89-91 and lines 326-328), while devoting a huge portion of the text to discussing stability vs maneuverability. The authors could have put their research into the context of the swimming literature over the last century instead of focusing on the last decade, to increase the broad interest of this study.

During the previous round of reviews, all three reviewers took issue with the authors' handling of the "stability vs maneuverability" issue. They point out that (1) the authors misrepresent the previous work in this area and (2) the authors' present study is limited in its ability to draw conclusions about stability and maneuverability, given the lack of data on dynamic control of maneuvering in boxfishes. I feel that the authors have made good progress towards addressing both of these issues in their revision, but I have a few suggestions to further improve the manuscript. As mentioned above, the authors could present the study in the broader context of the role of the caudal fin in fish swimming (especially steering vs. propulsion) and how boxfishes are unique. This would allow the authors to put their results into a much broader context of the fish swimming literature. As the other reviewers point out, little can be said about stability and maneuverability from this dataset, but the use of the caudal fin as a rudder is clearly demonstrated by this study.

In terms of addressing the previous work of Bartol and colleagues, my primary issue is with the repeated assertion that the keel-induced vortices of the boxfish carapace have a "negligible effect" on yaw. Bartol observed these vortices and found them to be important in self-correcting, especially for pitch but also for yaw. He did not assert that the entire carapace had a net stabilizing effect. Therefore, the "negligible effect" of these vortices documented by Van Wassenbergh et al. (2015) is not a direct rejection of Bartol's findings. Also, although Van Wassenbergh characterizes the effect of the vortices as "negligible" in his 2015 study, his graphs of yaw and pitch moments do show a stabilizing effect of the keel vortices. While the stabilizing moments were certainly less than the destabilizing moments, they were nowhere close to zero. In fact, Van Wassenbergh et al. (2015) acknowledges that they "could confirm the previously predicted stabilizing effect on the lateral carapace walls at the posterior end of the fish." However, no statistical analysis or evaluation was conducted to support the assertion that the stabilizing moments can be accurately described as "negligible" relative to the destabilizing moments. Furthermore, as the authors point out, none of the previous studies have included the large and distinctive boxfish scales, which likely have a large impact on flow separation and could drastically change modeling results. Therefore, the rejection of self-correcting keel vortices as "negligible" seems premature without additional work. It is outside the scope of the present study to address this, but I strongly encourage the authors to consider presenting a more balanced perspective on the literature. It is especially important for the authors to be cautious here, because they did not make direct observations of vortices, nor did they study how these vortices might interact with the caudal fin, despite mentioning this in the introduction.

In terms of technical details, I believe that the authors have adequately addressed previous issues with this paper. My only remaining technical concern is that I would like to see more information on the properties of the PMMA tail. Considering that this paper relies only on models, the description of "this material is relatively stiff but still bends a bit at larger forces" is not adequate. Did the material bend at the forces experienced at the highest flow speeds? Was any deformation observed during the experiment? Were fluid-solid interactions included in the computational model to address bending of the caudal fin? My general understanding is that fishes use muscles to control the shape of the tail and resist passive bending due to incoming flow, and therefore an inflexible material may have been more appropriate, unless the authors have previously observed passive bending of the caudal fin in boxfishes. Again, this is where observations of live animals are necessary and useful.

Minor comments:

The authors do a relatively good job of using "boxfishes" when generalizing about multiple species, but this should be checked throughout (see "boxfish" on lines 21, 29, 64, 70, 77, etc).

When discussing the choice of which flow speeds to use, the authors focus on reported fish swimming speeds. However, isn't the focus of this study on destabilizing incoming flows that might be experienced in a complex reef environment? How do the flow speeds reflect those kinds of interactions?

Lines 40-41 - The uses of the "/" symbol to indicate "and/or" are not appropriate. The terms "adapted," "specialised," "swimming modes," and "functions" have distinct definitions and cannot be used in an "and/or" context. The authors should choose the most accurate term in both cases or elaborate on the intended meaning. I encourage the authors to avoid the "and/or" construct throughout the rest of the paper.

Line 51 - Should read "ostraciid"

Lines 97-110 - It would be helpful if the authors would outline their many experiments explicitly in this paragraph, to provide a road map for the readers for this complex experimental design. For example, there is no mention of the tailless experiments in this paragraph.

Line 436 - The authors should also include STL or OBJ files for their 3D models, which are necessary for replication.

Review form: Reviewer 2

Is the manuscript scientifically sound in its present form?

Yes

Are the interpretations and conclusions justified by the results?

Yes

Is the language acceptable?

Yes

Do you have any ethical concerns with this paper?

No

Have you any concerns about statistical analyses in this paper?

No

Recommendation?

Major revision is needed (please make suggestions in comments)

Comments to the Author(s)

Overall Comments:

This manuscript investigates the effect of body angle and tail angle on the generation of yawing torques in a physical model of boxfishes. The major findings are as follows: (1) Yaw about a rigid body is generally destabilizing when the body has a non-zero angle of attack relative to incoming flow. (2) When a flexible caudal fin (though not one tuned to the properties of a biological fin) is added to the rigid model, the configuration becomes self stabilizing at increasing angles of attack. (3) Including a tail angle can create self-stabilizing yaw torques (negative yaw torques at positive body angles of attack).

While the modeling done for this study is mostly sound, I have specific concerns about the position of the sting and its effects on yawing moments; the lack of information about the flexural stiffness of the caudal fin model; and the use of the tail-less model as a kind of null for the hypothesis of self-stability. In addition, I question how informative this research is over previous studies. We already know that fins induce yawing moments. It does not seem like a notable

advance to quantify those moments in a model that may not match biological parameters observed in fishes (like tail stiffness).

Specific Comments:

Title: The title seems misleading here. This paper, to me, seems to be looking at modulation of passive stability in yaw, not maneuverability. In other words, I think of maneuverability as what the living fish will actually do to generate torques (actual behavior), not what it is hypothetically capable of doing.

Ln 54-56: The first half of this paragraph can be considered relevant context in that the armor plating of boxfishes inevitably will contribute to the range of motion of the body – but the discussion of ostracitoxin seems irrelevant in this context.

Ln 67-69: Suggest removal of the “the intended change...in a controlled manner”.

Ln 97-110: These should be in the past tense.

Ln 137-140: Why not place the sting at the center of mass, if its position in the fish specimens was known? This seems an avoidable source of error. I am concerned that even small differences in the position of the sting may have consequences for the location of the “switch points” discussed later on.

Ln 157: Does the caudal fin scale isometrically in the fish? Absence of studies of scaling does not mean evidence of isometry.

Ln 161-164: The flexural stiffness of the PMMA, or at least the Young's modulus, could be compared to the measured values for actual fishes. (e.g. Tangorra et al. 2010)

Ln 319: But others have found self-stabilizing flows over the carapace of boxfishes when including the caudal fin. It seems like a straw-man argument to draw “adaptive” conclusions about the relative stability of the boxfish carapace when it evolved in the context of having a caudal fin.

Tangorra JL, Lauder GV, Hunter IW, Mittal R, Madden PGA, Bozkurttas M. 2010 The effect of fin ray flexural rigidity on the propulsive forces generated by a biorobotic fish pectoral fin. *Journal of Experimental Biology* 213, 4043–4054. (doi:10.1242/jeb.048017)

Decision letter (RSOS-191335.R0)

16-Sep-2019

Dear Mr Boute,

The editors assigned to your paper ("Modulation of manoeuvrability with an unstable rigid body and a stabilising caudal fin in the yellow boxfish (*Ostracion cubicus*)") have now received comments from reviewers. We would like you to revise your paper in accordance with the referee and Associate Editor suggestions which can be found below (not including confidential reports to the Editor). Please note this decision does not guarantee eventual acceptance.

Please submit a copy of your revised paper before 09-Oct-2019. Please note that the revision deadline will expire at 00.00am on this date. If we do not hear from you within this time then it will be assumed that the paper has been withdrawn. In exceptional circumstances, extensions

may be possible if agreed with the Editorial Office in advance. We do not allow multiple rounds of revision so we urge you to make every effort to fully address all of the comments at this stage. If deemed necessary by the Editors, your manuscript will be sent back to one or more of the original reviewers for assessment. If the original reviewers are not available, we may invite new reviewers.

- Data accessibility

If you wish to submit your supporting data or code to Dryad (<http://datadryad.org/>), or modify your current submission to dryad, please use the following link:
<http://datadryad.org/submit?journalID=RSOS&manu=RSOS-191335>

- Competing interests

- Authors' contributions

- Acknowledgements

- Funding statement

Kind regards,

Andrew Dunn

on behalf of Professor Brooke Flammang (Associate Editor) and Kevin Padian (Subject Editor)

Associate Editor's comments (Professor Brooke Flammang):

Associate Editor: 1

Comments to the Author:

Both reviewers have provided supportive feedback that should guide the authors in improving the manuscript. In short, there is concern regarding (a) the biological relevance of the model, and (b) the novelty and broad interest of these findings. With regards to biological relevance, the authors state that they have access to live animals (although do not give animal care ethics and protocol information), yet they do not appear to base their model on live boxfish kinematics of the material properties of the tail. Thus, the assumptions drawn from the model are not validated in any way. With regards to the novelty and broad interest, a large amount of discussion revolves around stability and maneuverability, but as both reviewers point out, there is a body of work on this topic that this paper does not expand upon (and in some cases does not adequately describe). In fact, this manuscript may miss the mark on more relevant topics such as stabilizing hydrodynamics (which are not tested here but include some misstated assumptions as facts).

Editor comments:

As you can see the reviewers like the manuscript but they still have substantial concerns. If in your revision you feel you need more than the allotted time, please don't hesitate to contact the editorial office. Best wishes for your revision.

Comments to Author:

Reviewers' Comments to Author:

Reviewer: 1

Comments to the Author(s)

This article about the hydrodynamics of the caudal fin of the yellow boxfish is an interesting study that offers some insights into the utility of the large caudal fin of boxfishes, in coordination

with the rigid carapace. Using models, the authors make a compelling case that the caudal fin could act as a stabilizer in response to incoming flow.

The revisions made by the authors based on feedback from previous reviewers have substantially improved this manuscript. My concerns are primarily about the framing of this paper, both in terms of broadening its appeal for this journal and in terms of adequately putting the results into context.

Major comments:

I feel that the biggest weakness of this study is that there have been so few studies on the use of the caudal fin in living boxfishes. Detailed descriptions of when and how boxfishes use their caudal fins are necessary to interpret the modeling results of this study. This is a principle that applies to all modeling studies - while models are very informative, they are best interpreted in the context of how the animal is using the structure in life. For example, the authors show clear differences in yaw torque between an open and closed caudal fins, but there is no documentation of when boxfishes use an open or closed caudal fin. The only live boxfish studies that have included analysis of caudal fin motions that I am aware of are Walker (2000) and Gordon et al. (2000), which the authors cite, but these studies did not document whether the fin was open or closed during turning or burst swimming.

I understand that it is likely outside the scope of this study for the authors to address this, but I feel that a lack of live animal observations is a limitation of this study that reduces its potential impact. The authors state that they have these animals in the lab, so it seems that a reasonable first step would have been to simply document how the tail is being used (open vs. closed caudal fin, tail position during steering vs. stabilizing, etc.) before proceeding to models. From watching videos from aquarists online, I could easily imagine that boxfishes are using their caudal fin as a rudder, exactly as the authors suggest (they are certainly not using it in the same way as a bluegill). However, without proper documentation of these behaviors, much of the interpretation in the discussion is speculation. Simple experiments could be done to destabilize a living boxfish with bursts of water and observe how the caudal fin is used to respond to this. Even video analysis of a free-swimming animal in a complicated flow environment could be informative.

I also feel that the authors limit themselves by using primarily a "stability vs. maneuverability" framework in their writing. There is a much bigger insight to be gained here, which is that we have been misinterpreting "ostraciiform" locomotion for years. Walker (2000) and Gordon et al. (2000) made this point by clearly demonstrating that the caudal fin is rarely used for propulsion except in short burst-and-coast behaviors, which is in contrast with Breder's classification of ostraciiform locomotion as being caudal-driven. Walker, Gordon, and others have set up an excellent framework for this study: there is a continuum among fishes of caudal fins being used as a primarily as a propulsor or primarily as a rudder, and boxfishes represent an extreme on this spectrum. So, how does the use of the caudal fin in boxfishes differ from that of other fishes? How are the fluid dynamics of steering fundamentally different from propulsion, and how are the boxfishes taking advantage of this? This is an issue of framing. The authors give a very slight nod to these ideas (lines 89-91 and lines 326-328), while devoting a huge portion of the text to discussing stability vs maneuverability. The authors could have put their research into the context of the swimming literature over the last century instead of focusing on the last decade, to increase the broad interest of this study.

During the previous round of reviews, all three reviewers took issue with the authors' handling of the "stability vs maneuverability" issue. They point out that (1) the authors misrepresent the previous work in this area and (2) the authors' present study is limited in its ability to draw conclusions about stability and maneuverability, given the lack of data on dynamic control of maneuvering in boxfishes. I feel that the authors have made good progress towards addressing both of these issues in their revision, but I have a few suggestions to further improve the manuscript. As mentioned above, the authors could present the study in the broader context of

the role of the caudal fin in fish swimming (especially steering vs. propulsion) and how boxfishes are unique. This would allow the authors to put their results into a much broader context of the fish swimming literature. As the other reviewers point out, little can be said about stability and maneuverability from this dataset, but the use of the caudal fin as a rudder is clearly demonstrated by this study.

In terms of addressing the previous work of Bartol and colleagues, my primary issue is with the repeated assertion that the keel-induced vortices of the boxfish carapace have a "negligible effect" on yaw. Bartol observed these vortices and found them to be important in self-correcting, especially for pitch but also for yaw. He did not assert that the entire carapace had a net stabilizing effect. Therefore, the "negligible effect" of these vortices documented by Van Wassenbergh et al. (2015) is not a direct rejection of Bartol's findings. Also, although Van Wassenbergh characterizes the effect of the vortices as "negligible" in his 2015 study, his graphs of yaw and pitch moments do show a stabilizing effect of the keel vortices. While the stabilizing moments were certainly less than the destabilizing moments, they were nowhere close to zero. In fact, Van Wassenbergh et al. (2015) acknowledges that they "could confirm the previously predicted stabilizing effect on the lateral carapace walls at the posterior end of the fish." However, no statistical analysis or evaluation was conducted to support the assertion that the stabilizing moments can be accurately described as "negligible" relative to the destabilizing moments. Furthermore, as the authors point out, none of the previous studies have included the large and distinctive boxfish scales, which likely have a large impact on flow separation and could drastically change modeling results. Therefore, the rejection of self-correcting keel vortices as "negligible" seems premature without additional work. It is outside the scope of the present study to address this, but I strongly encourage the authors to consider presenting a more balanced perspective on the literature. It is especially important for the authors to be cautious here, because they did not make direct observations of vortices, nor did they study how these vortices might interact with the caudal fin, despite mentioning this in the introduction.

In terms of technical details, I believe that the authors have adequately addressed previous issues with this paper. My only remaining technical concern is that I would like to see more information on the properties of the PMMA tail. Considering that this paper relies only on models, the description of "this material is relatively stiff but still bends a bit at larger forces" is not adequate. Did the material bend at the forces experienced at the highest flow speeds? Was any deformation observed during the experiment? Were fluid-solid interactions included in the computational model to address bending of the caudal fin? My general understanding is that fishes use muscles to control the shape of the tail and resist passive bending due to incoming flow, and therefore an inflexible material may have been more appropriate, unless the authors have previously observed passive bending of the caudal fin in boxfishes. Again, this is where observations of live animals are necessary and useful.

Minor comments:

The authors do a relatively good job of using "boxfishes" when generalizing about multiple species, but this should be checked throughout (see "boxfish" on lines 21, 29, 64, 70, 77, etc).

When discussing the choice of which flow speeds to use, the authors focus on reported fish swimming speeds. However, isn't the focus of this study on destabilizing incoming flows that might be experienced in a complex reef environment? How do the flow speeds reflect those kinds of interactions?

Lines 40-41 - The uses of the "/" symbol to indicate "and/or" are not appropriate. The terms "adapted," "specialised," "swimming modes," and "functions" have distinct definitions and cannot be used in an "and/or" context. The authors should choose the most accurate term in both cases or elaborate on the intended meaning. I encourage the authors to avoid the "and/or" construct throughout the rest of the paper.

Line 51 - Should read "ostraciid"

Lines 97-110 - It would be helpful if the authors would outline their many experiments explicitly in this paragraph, to provide a road map for the readers for this complex experimental design. For example, there is no mention of the tailless experiments in this paragraph.

Line 436 - The authors should also include STL or OBJ files for their 3D models, which are necessary for replication.

Reviewer: 2

Comments to the Author(s)

Overall Comments:

This manuscript investigates the effect of body angle and tail angle on the generation of yawing torques in a physical model of boxfishes. The major findings are as follows: (1) Yaw about a rigid body is generally destabilizing when the body has a non-zero angle of attack relative to incoming flow. (2) When a flexible caudal fin (though not one tuned to the properties of a biological fin) is added to the rigid model, the configuration becomes self stabilizing at increasing angles of attack. (3) Including a tail angle can create self-stabilizing yaw torques (negative yaw torques at positive body angles of attack).

While the modeling done for this study is mostly sound, I have specific concerns about the position of the sting and its effects on yawing moments; the lack of information about the flexural stiffness of the caudal fin model; and the use of the tail-less model as a kind of null for the hypothesis of self-stability. In addition, I question how informative this research is over previous studies. We already know that fins induce yawing moments. It does not seem like a notable advance to quantify those moments in a model that may not match biological parameters observed in fishes (like tail stiffness).

Specific Comments:

Title: The title seems misleading here. This paper, to me, seems to be looking at modulation of passive stability in yaw, not maneuverability. In other words, I think of maneuverability as what the living fish will actually do to generate torques (actual behavior), not what it is hypothetically capable of doing.

Ln 54-56: The first half of this paragraph can be considered relevant context in that the armor plating of boxfishes inevitably will contribute to the range of motion of the body - but the discussion of ostracitoxin seems irrelevant in this context.

Ln 67-69: Suggest removal of the "the intended change...in a controlled manner".

Ln 97-110: These should be in the past tense.

Ln 137-140: Why not place the sting at the center of mass, if its position in the fish specimens was known? This seems an avoidable source of error. I am concerned that even small differences in the position of the sting may have consequences for the location of the "switch points" discussed later on.

Ln 157: Does the caudal fin scale isometrically in the fish? Absence of studies of scaling does not mean evidence of isometry.

Ln 161-164: The flexural stiffness of the PMMA, or at least the Youngs modulus, could be compared to the measured values for actual fishes. (e.g. Tangorra et al. 2010)

Ln 319: But others have found self-stabilizing flows over the carapace of boxfishes when including the caudal fin. It seems like a straw-man argument to draw “adaptive” conclusions about the relative stability of the boxfish carapace when it evolved in the context of having a caudal fin.

Tangorra JL, Lauder GV, Hunter IW, Mittal R, Madden PGA, Bozkurttas M. 2010 The effect of fin ray flexural rigidity on the propulsive forces generated by a biorobotic fish pectoral fin. *Journal of Experimental Biology* 213, 4043–4054. (doi:10.1242/jeb.048017)

Author's Response to Decision Letter for (RSOS-191335.R0)

See Appendix A.

RSOS-191335.R1 (Revision)

Review form: Reviewer 1

Is the manuscript scientifically sound in its present form?

Yes

Are the interpretations and conclusions justified by the results?

Yes

Is the language acceptable?

Yes

Do you have any ethical concerns with this paper?

No

Have you any concerns about statistical analyses in this paper?

No

Recommendation?

Accept with minor revision (please list in comments)

Comments to the Author(s)

This article about the hydrodynamics of the caudal fin of the yellow boxfish is an interesting study that offers some insights into the utility of the large caudal fin of boxfishes, in coordination with the rigid carapace. Using models, the authors make a compelling case that the caudal fin could act as a stabilizer in response to incoming flow.

The authors did an adequate job with this revision. They added some crucial information in support of their study. I still have a few concerns, described below. However, the authors have made their decisions in terms of how to frame the study and how to address the literature. It is now up to the editors to decide if the article is suitable for their audience and meets their standards of literature review. My assessment is that the article is scientifically sound and

suitable for publication, in addition to providing valuable information about boxfish swimming hydrodynamics.

A major improvement to the manuscript is the addition of information about the position of the caudal fin in live boxfishes from the literature. The authors provide details of Blake's observations of boxfish caudal fins (i.e., the behaviors during which the fin is open or closed and the degree to which the fin is angled). This context provides support for the realism of this model. Previously, the article was written as though there was no documentation of when and how an open or closed caudal fin was used by boxfishes, which made it difficult to evaluate the utility of this model. I encourage the authors to incorporate a summary of Blake's observations into the introduction, because they are crucial to interpreting the experiments. The current statement (lines 92-95) is too vague.

I am glad that the authors have provided a Young's modulus for the PMMA material. However, the authors did not address my questions from the previous review: "Did the material bend at the forces experienced at the highest flow speeds? Was any deformation observed during the experiment?" Since the authors state that the material "is relatively stiff but still bends a bit at larger forces," I would like to know if any bending was observed. Considerable bending could have a significant impact on interpretation of the results. If the authors observed minimal to no bending, this should be reported to address this concern.

I still feel that the "stability vs maneuverability" framework does not serve this paper well. I feel that the authors have demonstrated the use of the caudal fin as a rudder for steering, as the authors repeatedly state. However, the utility of the caudal fin in dynamic control of course stability is still unknown and is, as the authors state, outside the scope of this study. However, the authors have made their case for this framing.

It seems unnecessary and inappropriate to repeatedly reject the findings of previous studies of carapace hydrodynamics, as opposed to building upon and updating previous understandings. As many previous referees have pointed out, the authors could provide a more balanced and nuanced discussion rather than outright rejection of the work by Bartol and colleagues.

I find no evidence in any of the Bartol articles or in Farina and Summers (2015) that Bartol and colleagues asserted that the entire carapace had a net stabilizing effect, despite the authors repeatedly making this claim in response to reviewers. As Farina and Summers point out, Bartol and colleagues focus on their discovery of stabilizing vortices associated with the keels, and they provide ample evidence of this. While Van Wassenbergh (2015) provides strong evidence of a net destabilizing effect of carapace hydrodynamics as a whole, which is a major contribution to our understanding of boxfish biology, this does not refute the presence of the keel vortices or their potential to provide stabilization in some contexts (either in biomimetic design or in the biology of the animals). More research is clearly needed. As Farina and Summers point out (and as summarized in their figure), Van Wassenbergh's study provides strong evidence against the self-stabilizing nature of the entire carapace, but that does not mean that the entirety of Bartol and colleagues previous work on keel vortices is invalidated.

Also, as a side note, while I agree that Farina and Summers provide a balanced perspective, it may not be appropriate to cite a News and Views article in this context. On line 395, it is cited as if it were a research article, which seems a bit misleading.

As for data accessibility, is there something that you can do to reduce the file size for download from DataDryad? 40 GB is a lot. Perhaps files could be grouped and stored as separate DataDryad entries, so that not all files need to be downloaded at once.

Review form: Reviewer 2

Is the manuscript scientifically sound in its present form?

Yes

Are the interpretations and conclusions justified by the results?

Yes

Is the language acceptable?

Yes

Do you have any ethical concerns with this paper?

No

Have you any concerns about statistical analyses in this paper?

No

Recommendation?

Major revision is needed (please make suggestions in comments)

Comments to the Author(s)

While previous reviewers have provided a wealth of feedback to improve this manuscript, the authors have by and large chosen to rebut this feedback, and reassert some of their more problematic claims. As a result, I am hesitant to recommend this manuscript for publication.

A previous reviewer suggested reframing in the context of the historical literature (ostraciiform locomotion writ large) that would alleviate much of this issue. In short, my concerns with this manuscript are less to do with the experiments, and more to do with the interpretation of the data in the context of the literature.

The authors maintain that both the reviewers on this manuscript, and those of the previous (2015) manuscript have incorrect assertions about the literature regarding stability and maneuverability. This is deeply troubling: if that many professionals in swimming biomechanics (the reviewers) disagree with the authors' interpretation, then the authors need to do a better job justifying their interpretation, beyond citing their own paper. The authors suggest that Farina and Summers provide an "independent assessment" of this topic, but the article they reference is merely a summary of their article, and not an independent assessment. This manuscript would be made much stronger by a more balanced perspective on the literature, but the authors do not appear willing to take this approach.

Decision letter (RSOS-191335.R1)

18-Dec-2019

Dear Mr Boute:

Manuscript ID RSOS-191335.R1 entitled "Modulation of yaw stability with an unstable rigid body and a stabilising caudal fin in the yellow boxfish (*Ostracion cubicus*)" which you submitted to Royal Society Open Science, has been reviewed. The comments from reviewer(s) are included at the bottom of this letter.

In view of the criticisms of the reviewer(s), I must decline the manuscript for publication in Royal

Society Open Science at this time. However, a new manuscript may be submitted which takes into consideration these comments.

Please note that resubmitting your manuscript does not guarantee eventual acceptance, and that your resubmission will be subject to re-review by the reviewer(s) before a decision is rendered.

You will be unable to make your revisions on the originally submitted version of your manuscript. Instead, revise your manuscript using a word processing program and save it on your computer.

You may also click the below link to start the resubmission process (or continue the process if you have already started your resubmission) for your manuscript. If you use the below link you will not be required to login to ScholarOne Manuscripts.

*** PLEASE NOTE: This is a two-step process. After clicking on the link, you will be directed to a webpage to confirm. ***

https://mc.manuscriptcentral.com/rsos?URL_MASK=3290c7a83c554b988aa93c4aa3270ba3

Because we are trying to facilitate timely publication of manuscripts submitted to Royal Society Open Science, your resubmitted manuscript should be submitted by 16-Jun-2020. If you are unable to submit by this date please contact the Editorial Office for options.

I look forward to a resubmission.

on behalf of Professor Brooke Flammang (Associate Editor) and Kevin Padian (Subject Editor)
openscience@royalsociety.org

Associate Editor Comments to Author (Professor Brooke Flammang):

Both reviewers have seen that the authors have made some efforts to revise their paper. However, of a more major concern is that there are some important issues with regards to arguments laying foundation for the major findings of this paper that have been rebutted in the response instead of clarified within the manuscript itself. I encourage the authors to consider that whether they think the reviewers are more correct in their interpretation or not, it is the role of the author is to provide a clear explanation of the science being discussed. If several reviewers have brought up similar issues disagreeing with the authors' point of view, it seems that the explanation provided in the manuscript is perhaps not as clear as it could be.

Subject Editor comments:

Thanks for your revisions. It is very important to lay out the work and arguments of previous research clearly, and our reviewers and AE think that you have not yet brought the manuscript to

that point. Please provide your best explanation in your revision. We will be unable to send it out for another round otherwise. Thanks again.

Reviewer comments to Author:

Reviewer: 1

Comments to the Author(s)

This article about the hydrodynamics of the caudal fin of the yellow boxfish is an interesting study that offers some insights into the utility of the large caudal fin of boxfishes, in coordination with the rigid carapace. Using models, the authors make a compelling case that the caudal fin could act as a stabilizer in response to incoming flow.

The authors did an adequate job with this revision. They added some crucial information in support of their study. I still have a few concerns, described below. However, the authors have made their decisions in terms of how to frame the study and how to address the literature. It is now up to the editors to decide if the article is suitable for their audience and meets their standards of literature review. My assessment is that the article is scientifically sound and suitable for publication, in addition to providing valuable information about boxfish swimming hydrodynamics.

A major improvement to the manuscript is the addition of information about the position of the caudal fin in live boxfishes from the literature. The authors provide details of Blake's observations of boxfish caudal fins (i.e., the behaviors during which the fin is open or closed and the degree to which the fin is angled). This context provides support for the realism of this model. Previously, the article was written as though there was no documentation of when and how an open or closed caudal fin was used by boxfishes, which made it difficult to evaluate the utility of this model. I encourage the authors to incorporate a summary of Blake's observations into the introduction, because they are crucial to interpreting the experiments. The current statement (lines 92-95) is too vague.

I am glad that the authors have provided a Young's modulus for the PMMA material. However, the authors did not address my questions from the previous review: "Did the material bend at the forces experienced at the highest flow speeds? Was any deformation observed during the experiment?" Since the authors state that the material "is relatively stiff but still bends a bit at larger forces," I would like to know if any bending was observed. Considerable bending could have a significant impact on interpretation of the results. If the authors observed minimal to no bending, this should be reported to address this concern.

I still feel that the "stability vs maneuverability" framework does not serve this paper well. I feel that the authors have demonstrated the use of the caudal fin as a rudder for steering, as the authors repeatedly state. However, the utility of the caudal fin in dynamic control of course stability is still unknown and is, as the authors state, outside the scope of this study. However, the authors have made their case for this framing.

It seems unnecessary and inappropriate to repeatedly reject the findings of previous studies of carapace hydrodynamics, as opposed to building upon and updating previous understandings. As many previous referees have pointed out, the authors could provide a more balanced and nuanced discussion rather than outright rejection of the work by Bartol and colleagues.

I find no evidence in any of the Bartol articles or in Farina and Summers (2015) that Bartol and colleagues asserted that the entire carapace had a net stabilizing effect, despite the authors repeatedly making this claim in response to reviewers. As Farina and Summers point out, Bartol and colleagues focus on their discovery of stabilizing vortices associated with the keels, and they provide ample evidence of this. While Van Wassenbergh (2015) provides strong evidence of a net

destabilizing effect of carapace hydrodynamics as a whole, which is a major contribution to our understanding of boxfish biology, this does not refute the presence of the keel vortices or their potential to provide stabilization in some contexts (either in biomimetic design or in the biology of the animals). More research is clearly needed. As Farina and Summers point out (and as summarized in their figure), Van Wassenbergh's study provides strong evidence against the self-stabilizing nature of the entire carapace, but that does not mean that the entirety of Bartol and colleagues previous work on keel vortices is invalidated.

Also, as a side note, while I agree that Farina and Summers provide a balanced perspective, it may not be appropriate to cite a News and Views article in this context. On line 395, it is cited as if it were a research article, which seems a bit misleading.

As for data accessibility, is there something that you can do to reduce the file size for download from DataDryad? 40 GB is a lot. Perhaps files could be grouped and stored as separate DataDryad entries, so that not all files need to be downloaded at once.

Reviewer: 2

Comments to the Author(s)

While previous reviewers have provided a wealth of feedback to improve this manuscript, the authors have by and large chosen to rebut this feedback, and reassert some of their more problematic claims. As a result, I am hesitant to recommend this manuscript for publication.

A previous reviewer suggested reframing in the context of the historical literature (ostraciiform locomotion writ large) that would alleviate much of this issue. In short, my concerns with this manuscript are less to do with the experiments, and more to do with the interpretation of the data in the context of the literature.

The authors maintain that both the reviewers on this manuscript, and those of the previous (2015) manuscript have incorrect assertions about the literature regarding stability and maneuverability. This is deeply troubling: if that many professionals in swimming biomechanics (the reviewers) disagree with the authors' interpretation, then the authors need to do a better job justifying their interpretation, beyond citing their own paper. The authors suggest that Farina and Summers provide an "independent assessment" of this topic, but the article they reference is merely a summary of their article, and not an independent assessment. This manuscript would be made much stronger by a more balanced perspective on the literature, but the authors do not appear willing to take this approach.

Author's Response to Decision Letter for (RSOS-191335.R1)

See Appendix B.

RSOS-200129.R0

Review form: Reviewer 1

Is the manuscript scientifically sound in its present form?

Yes

Are the interpretations and conclusions justified by the results?

Yes

Is the language acceptable?

Yes

Do you have any ethical concerns with this paper?

No

Have you any concerns about statistical analyses in this paper?

No

Recommendation?

Accept as is

Comments to the Author(s)

The authors have adequately responded to all of my concerns, and I am satisfied with this revision. I still have a minor concern, but the authors can decide whether they want to address it, and there is no need for me to see another revision.

In my last set of comments, one of my concerns was that the authors still characterized the work by Bartol and colleagues to have concluded that the shape of the carapace had a net-stabilizing effect. However, there is confusion about "net stabilizing" vs "self-correcting." Bartol et al certainly repeatedly claimed that the vortices were self-correcting and self-stabilizing, and Van Wassenbergh shows that the carapace as a whole had net-destabilizing tendencies, a subtle distinction that was summarized briefly in the Farina and Summers review (in particular, see their figure). However, I'm still not convinced that their goal was ever to show that the entire carapace was "net stabilizing," which is evidenced by the fact that Bartol and colleagues never state this outright and only rarely measured net torque (and even then only in pitch, not in yaw).

The authors have done a nice job revising how they address the Bartol et al experiments, but the paragraph where they summarize this (new lines 70-81) still sets up a framework of: (A) A hypothesis of self-correcting vortices was proposed by Bartol et al and (B) Van Wassenbergh et al rejected that hypothesis because they found the carapace to net-stabilizing. If a net-stabilizing carapace and self-correcting vortices are not mutually-exclusive hypotheses, I still don't think this framework is appropriate.

I leave it up to the authors and editors to decide if this point is important enough to warrant further revision.

I appreciate the opportunity to review this manuscript, and I congratulate the authors on a nice study that will advance the field.

Review form: Reviewer 3

Is the manuscript scientifically sound in its present form?

Yes

Are the interpretations and conclusions justified by the results?

No

Is the language acceptable?

Yes

Do you have any ethical concerns with this paper?

No

Have you any concerns about statistical analyses in this paper?

No

Recommendation?

Major revision is needed (please make suggestions in comments)

Comments to the Author(s)

Review comments on RSOS-200129

In this study, the authors investigated the yaw control of the boxfish by caudal fin. They demonstrated that the rigid body was unstable, while the caudal fin acted as both a stabilizer and rudder through an interplay with the rigid body. The data in this study is valuable, but I think the interpretation on the data ought to be improved.

1. The title only focuses on the 'stability' role of the caudal fin, but the manuscript actually tells that the caudal fin is both a stabilizer and rudder (stability and maneuverability). Hence, the title seems inaccurate and providing a lopsided first impression.

2. My major concern on the authors' interpretation on their data is: what is stability/stabilizing? The authors seem considering this issue simply: when an including angle occurs between fish and oncoming flow (body angle α in the text), effects reducing this angle are stabilizing, while effects increasing this angle are destabilizing.

However, we may imagine the following two circumstances: First, the boxfish is still and a gust flow comes laterally; Second, the boxfish is swimming forward while a gust flow comes laterally. In both circumstances, an angle occurs between fish and oncoming flow (for swimming condition, oncoming flow speed is the composite speed of swimming speed and gust flow speed). Neither increment nor decrement of this angle brings stability. Instead, stability means maintaining the current heading direction of fish, which corresponds to the ZERO torque points in Fig.7.

Minor comments:

3. Line 70-81, in this paragraph, the authors actually list two hypotheses of the previous studies. First hypothesis: carapace induces stabilizing vortices; second hypothesis: carapace-induced stabilizing vortices may lead to overall stability. Rather than stating 'THIS hypothesis was rejected', the authors would better improve their description to clarify which hypothesis was rejected.

4. Could the authors explain those large errors of the data points at 40 degrees in Fig.5(b)?

5. Line 402, since the reference Farina & Summers [45] is not a research, but a news report based on the reference Van Wassenbergh [11], it should not appear in the discussion. If the authors think it's important, I suggest move it to somewhere in the introduction part, with highlight of the type of this reference.

Decision letter (RSOS-200129.R0)

27-Feb-2020

Dear Mr Boute

On behalf of the Editor, I am pleased to inform you that your Manuscript RSOS-200129 entitled "Modulation of yaw stability with an unstable rigid body and a stabilising caudal fin in the yellow boxfish (*Ostracion cubicus*)" has been accepted for publication in Royal Society Open Science subject to minor revision in accordance with the referee suggestions. Please find the referees' comments at the end of this email.

The reviewers and Subject Editor have recommended publication, but also suggest some minor revisions to your manuscript. Therefore, I invite you to respond to the comments and revise your manuscript.

- Ethics statement

- Data accessibility

<http://datadryad.org/submit?journalID=RSOS&manu=RSOS-200129>

- Competing interests

- Authors' contributions

AB carried out the molecular lab work, participated in data analysis, carried out sequence alignments, participated in the design of the study and drafted the manuscript; CD carried out the statistical analyses; EF collected field data; GH conceived of the study, designed the study,

coordinated the study and helped draft the manuscript. All authors gave final approval for publication.

- Acknowledgements

- Funding statement

Because the schedule for publication is very tight, it is a condition of publication that you submit the revised version of your manuscript before 07-Mar-2020. Please note that the revision deadline will expire at 00.00am on this date. If you do not think you will be able to meet this date please let me know immediately.

on behalf of Professor Brooke Flammang (Associate Editor) and Kevin Padian (Subject Editor)
openscience@royalsociety.org

Subject Editor Comments to Author (Professor Kevin Padian):

Comments to the Author:

Dear authors, we will be happy to accept your manuscript but I would like you to consider and respond to the comments of the single reviewer who still has reservations when you resubmit your manuscript in final form. Thanks very much.

Reviewer comments to Author:

Reviewer: 1

Comments to the Author(s)

The authors have adequately responded to all of my concerns, and I am satisfied with this revision. I still have a minor concern, but the authors can decide whether they want to address it, and there is no need for me to see another revision.

In my last set of comments, one of my concerns was that the authors still characterized the work by Bartol and colleagues to have concluded that the shape of the carapace had a net-stabilizing effect. However, there is confusion about "net stabilizing" vs "self-correcting." Bartol et al certainly repeatedly claimed that the vortices were self-correcting and self-stabilizing, and Van Wassenbergh shows that the carapace as a whole had net-destabilizing tendencies, a subtle distinction that was summarized briefly in the Farina and Summers review (in particular, see their figure). However, I'm still not convinced that their goal was ever to show that the entire carapace was "net stabilizing," which is evidenced by the fact that Bartol and colleagues never state this outright and only rarely measured net torque (and even then only in pitch, not in yaw).

The authors have done a nice job revising how they address the Bartol et al experiments, but the paragraph where they summarize this (new lines 70-81) still sets up a framework of: (A) A hypothesis of self-correcting vortices was proposed by Bartol et al and (B) Van Wassenbergh et al rejected that hypothesis because they found the carapace to net-stabilizing. If a net-stabilizing carapace and self-correcting vortices are not mutually-exclusive hypotheses, I still don't think this framework is appropriate.

I leave it up to the authors and editors to decide if this point is important enough to warrant further revision.

I appreciate the opportunity to review this manuscript, and I congratulate the authors on a nice study that will advance the field.

Reviewer: 3

Comments to the Author(s)

Review comments on RSOS-200129

In this study, the authors investigated the yaw control of the boxfish by caudal fin. They demonstrated that the rigid body was unstable, while the caudal fin acted as both a stabilizer and rudder through an interplay with the rigid body. The data in this study is valuable, but I think the interpretation on the data ought to be improved.

1. The title only focuses on the 'stability' role of the caudal fin, but the manuscript actually tells that the caudal fin is both a stabilizer and rudder (stability and maneuverability). Hence, the title seems inaccurate and providing a lopsided first impression.

2. My major concern on the authors' interpretation on their data is: what is stability/stabilizing? The authors seem considering this issue simply: when an including angle occurs between fish and oncoming flow (body angle α in the text), effects reducing this angle are stabilizing, while effects increasing this angle are destabilizing.

However, we may imagine the following two circumstances: First, the boxfish is still and a gust flow comes laterally; Second, the boxfish is swimming forward while a gust flow comes laterally. In both circumstances, an angle occurs between fish and oncoming flow (for swimming condition, oncoming flow speed is the composite speed of swimming speed and gust flow speed). Neither increment nor decrement of this angle brings stability. Instead, stability means maintaining the current heading direction of fish, which corresponds to the ZERO torque points in Fig.7.

Minor comments:

3. Line 70-81, in this paragraph, the authors actually list two hypotheses of the previous studies. First hypothesis: carapace induces stabilizing vortices; second hypothesis: carapace-induced stabilizing vortices may lead to overall stability. Rather than stating 'THIS hypothesis was rejected', the authors would better improve their description to clarify which hypothesis was rejected.

4. Could the authors explain those large errors of the data points at 40 degrees in Fig.5(b)?

5. Line 402, since the reference Farina & Summers [45] is not a research, but a news report based on the reference Van Wassenbergh [11], it should not appear in the discussion. If the authors think it's important, I suggest move it to somewhere in the introduction part, with highlight of the type of this reference.

Author's Response to Decision Letter for (RSOS-200129.R0)

See Appendix C.

Decision letter (RSOS-200129.R1)

16-Mar-2020

Dear Mr Boute,

It is a pleasure to accept your manuscript entitled "Modulating yaw with an unstable rigid body

and a course-stabilising or steering caudal fin in the yellow boxfish (*Ostracion cubicus*)" in its current form for publication in Royal Society Open Science. The comments of the reviewer(s) who reviewed your manuscript are included at the foot of this letter.

on behalf of Professor Brooke Flammang (Associate Editor) and Kevin Padian (Subject Editor)
openscience@royalsociety.org

Associate Editor Comments to Author (Professor Brooke Flammang):

Comments to the Author:

I appreciate that the authors have taken the reviewers comments in earnest and made the requested clarifications in text.

Appendix A

We thank the reviewers and editor for their valuable suggestions on how to improve this article. We appreciate their time and effort for making these reviews. We hope that our revision of the manuscript and clarifications in this response-to-referees letter make it suitable for publication in the Royal Society Open Science.

In the text below, the original review reports are given, with our replies printed in bold after an indented arrow. All changes to the text are printed in orange in the uploaded manuscript file. In addition, we also uploaded a 'clean' version.

Associate Editor's comments (Professor Brooke Flammang):

Associate Editor: 1

Comments to the Author:

Both reviewers have provided supportive feedback that should guide the authors in improving the manuscript. In short, there is concern regarding (a) the biological relevance of the model, and (b) the novelty and broad interest of these findings. With regards to biological relevance, the authors state that they have access to live animals (although do not give animal care ethics and protocol information), yet they do not appear to base their model on live boxfish kinematics of the material properties of the tail. Thus, the assumptions drawn from the model are not validated in any way. With regards to the novelty and broad interest, a large amount of discussion revolves around stability and maneuverability, but as both reviewers point out, there is a body of work on this topic that this paper does not expand upon (and in some cases does not adequately describe). In fact, this manuscript may miss the mark on more relevant topics such as stabilizing hydrodynamics (which are not tested here but include some misstated assumptions as facts).

- **We take note of this constructive criticism, and have done all we could to improve our manuscript. Yet, we also hope that the editor agrees that models do not necessarily become irrelevant if not all biological complexity is included. For example, do we need a caudal fin that incorporates truly realistic profiles due to fluid-structure interaction if the purpose is to test how yaw torque can be influenced by spreading and closing of the caudal fin, and turning the tail? One can even argue that the simplified models, that do not include interactions between e.g. caudal fin spreading and water-induced deformation to be more informative. By measuring a large 'morphospace' of potentially realisable shapes and postures, we highlight some of the major mechanical principles that are involved, but at the same time we acknowledge potential deviations for reality due to model simplifications. Note that in response to the previous round of revisions, we have added acknowledgements of potential effects that we did include in our study, e.g. regarding the caudal fin material: "though commonly observed types of morphing of the fins due to the interaction with the water, fin-ray geometry, and the fin's material properties, such as cupping (Zhu & Shoele, 2008), was not included in our simplified model."**
- **We indeed had access to live animals but filming the animals to reconstruct the caudal fin from still images is not considered to be an animal experiment according to the European Union's Directive on the protection of animals used for science.**
- **As you will see from our responses to the reviewers, misconceptions persist on prior literature (e.g. claims that 'Bartol and colleagues did not assert that the entire carapace had a net stabilizing effect' despite that this is exactly what they show in all their graphs reporting torques) which resulted in unjustified criticism on our previous publication, and on the discussion section of the current manuscript. This, and also the earlier discussion with the J R Soc Interface reviewers, seems to prove that a debate on stability and manoeuvrability still is important."**
- **We reduced clutter in figures 5, 6, and 7 by removing unnecessary y-axis and x-axis titles and labels.**

→ Regarding the “Data accessibility”: The IGES and CFD data files are permanently accessible through Dryad via the Dryad review URL https://datadryad.org/stash/share/Ap_vzG0my-BwRg3wolUA1RJge0CJsdURY0IX10AhZ7M and the Dryad DOI <https://doi.org/10.5061/dryad.dz08kprsv>.

Reference:

- Zhu, Q, Shoole K. 2008 Propulsion performance of skeleton-strengthened fin. *J. Exp. Biol.* 211, 2087-2100. (doi:10.1242/jeb.016279)

Editor comments:

As you can see the reviewers like the manuscript but they still have substantial concerns. If in your revision you feel you need more than the allotted time, please don't hesitate to contact the editorial office. Best wishes for your revision.

Comments to Author:

Reviewers' Comments to Author:

Reviewer: 1

Comments to the Author(s)

This article about the hydrodynamics of the caudal fin of the yellow boxfish is an interesting study that offers some insights into the utility of the large caudal fin of boxfishes, in coordination with the rigid carapace. Using models, the authors make a compelling case that the caudal fin could act as a stabilizer in response to incoming flow.

The revisions made by the authors based on feedback from previous reviewers have substantially improved this manuscript. My concerns are primarily about the framing of this paper, both in terms of broadening its appeal for this journal and in terms of adequately putting the results into context.

Major comments:

I feel that the biggest weakness of this study is that there have been so few studies on the use of the caudal fin in living boxfishes. Detailed descriptions of when and how boxfishes use their caudal fins are necessary to interpret the modeling results of this study. This is a principle that applies to all modeling studies - while models are very informative, they are best interpreted in the context of how the animal is using the structure in life. For example, the authors show clear differences in yaw torque between an open and closed caudal fins, but there is no documentation of when boxfishes use an open or closed caudal fin. The only live boxfish studies that have included analysis of caudal fin motions that I am aware of are Walker (2000) and Gordon et al. (2000), which the authors cite, but these studies did not document whether the fin was open or closed during turning or burst swimming.

→ We agree with the referee that, unfortunately, few studies have focused on the use of the caudal fin in living boxfishes. Walker (2000) and Gordon *et al.* (2000) indeed link posture of the caudal fin and body to turning in boxfish which we mention in the discussion for Walker: “Walker [20] studied movement of the tail in relation to the body of the white-spotted boxfish *Ostracion meleagris* and found that *O. meleagris* is partially controlling its turning kinematics by using the caudal fin as a rudder.” In the revision, this sentence, to fit the other changes as described below, was rewritten into: Walker [20] studied movement of the tail in relation to the body of the white-spotted boxfish *Ostracion meleagris* and found that the turning kinematics are partially controlled by using the caudal fin as a

rudder.” In addition, Blake (1977) also studied the position of the caudal fin in relation to yaw turning, which we mention in the discussion as well: “Blake [19] qualitatively linked posture and state of the caudal fin to yaw, roll and pitch movements for the longhorn cowfish *Lactoria cornuta* and the humpback turretfish *Tetrosomus gibbosus*.”

- We agree with the referee that documentation of when living boxfishes use a closed and open caudal fin is limited. However, Blake (1977) does provide information on the state of the caudal fin in *Lactoria cornuta* and *Tetrosomus gibbosus* and we added a sentence on this to the introduction of the manuscript: “In addition, the state of the caudal fin (e.g. completely closed or open) and angle with respect to the body axis is expected to be important during turning manoeuvres according to qualitative descriptions of living swimming boxfish specimens [13,19], and correlations found between the tail angle and turning radius [20].”
- Furthermore, we added additional information on this to the discussion of the manuscript: “During slow forward swimming, the caudal fin remains in a closed position and is aligned along the median axis whilst making small lateral excursions that counter lateral deflections of the body [19]. During fast forward swimming, the caudal fin is opened and moves laterally from the median plane in an arc of about 35° [19]. The state of the caudal fin was therefore hypothesised to contribute to swimming course stability [11], which is in line with our quantitative findings in static models of *O. cubicus*. The seemingly active lateral movements of the caudal fin during fast swimming as observed by Gordon [13] and Blake [19] indicates body/caudal fin propulsion as originally termed by Breder [28] as ostraciiform locomotion [68]. However, apart from fast burst-and-coast swimming, propulsion is median/paired fin driven, engaging the caudal fin as rudder [13,19,20,28]. This role of the caudal fin in swimming is in accordance with the large yaw torques generated due to the presence of the caudal fin as measured in the current study. Walker [20] studied movement of the tail in relation to the body of the white-spotted boxfish *Ostracion meleagris* and found that the turning kinematics are partially controlled by using the caudal fin as a rudder. During turning and rotation, the caudal fin is closed [19] and the tail rotates in opposite direction compared to the body [20], matching our findings.”

References:

- Blake RW. 1977 On ostraciiform locomotion. *J. Mar. Biol. Assoc. UK.* 57, 1047–1055. (doi:10.1017/S0025315400026114)
- Blake RW. 1981 Mechanics of ostraciiform propulsion. *Can. J. Zool.* 59, 1067–1071. (doi:10.1139/z81-148)
- Breder CM. 1926 The locomotion of fishes. *Zoologica* 4, 159–297.
- Gordon MS, Hove JR, Webb, PW, Weihs D. 2000 Boxfishes as unusually well-controlled autonomous underwater vehicles. *Physiol. Biochem. Zool.* 73, 663–671. (doi:10.1086/318098)
- Van Wassenbergh S, van Manen K, Marcroft TA, Alfaro ME, Stamhuis EJ. 2015 Boxfish swimming paradox resolved: forces by the flow of water around the body promote manoeuvrability. *J. R. Soc. Interface* 12, 20141146. (doi:10.1098/rsif.2014.1146)
- Walker JA. 2000 Does a rigid body limit maneuverability? *J. Exp. Biol.* 203, 3391–3396.

I understand that it is likely outside the scope of this study for the authors to address this, but I feel that a lack of live animal observations is a limitation of this study that reduces its potential impact. The authors state that they have these animals in the lab, so it seems that a reasonable first step would have been to simply document how the tail is being used (open vs. closed caudal fin, tail position during steering vs. stabilizing, etc.) before proceeding to models. From watching videos from aquarists online, I could easily imagine that boxfishes are using their caudal fin as a rudder, exactly as the authors suggest (they are certainly not using it in the same way as a bluegill).

However, without proper documentation of these behaviors, much of the interpretation in the discussion is speculation. Simple experiments could be done to destabilize a living boxfish with bursts of water and observe how the caudal fin is used to respond to this. Even video analysis of a free-swimming animal in a complicated flow environment could be informative.

→ **We agree with the referee that live animal observations would be a nice addition but this is indeed outside of the scope of this study. Nevertheless, Blake (1977) documented the tail and caudal fin movements and state (e.g. closed or open) in living specimens in two other boxfish species, namely the longhorn cowfish *Lactoria cornuta* and the humpback turretfish *Tetrosomus gibbosus*. We added additional information on this to the introduction and discussion of the manuscript as described above.**

References:

- **Blake RW. 1977 On ostraciiform locomotion. *J. Mar. Biol. Assoc. UK.* 57, 1047–1055. (doi:10.1017/S0025315400026114)**

I also feel that the authors limit themselves by using primarily a "stability vs. maneuverability" framework in their writing. There is a much bigger insight to be gained here, which is that we have been misinterpreting "ostraciiform" locomotion for years. Walker (2000) and Gordon et al. (2000) made this point by clearly demonstrating that the caudal fin is rarely used for propulsion except in short burst-and-coast behaviors, which is in contrast with Breder's classification of ostraciiform locomotion as being caudal-driven. Walker, Gordon, and others have set up an excellent framework for this study: there is a continuum among fishes of caudal fins being used as a primarily as a propulsor or primarily as a rudder, and boxfishes represent an extreme on this spectrum. So, how does the use of the caudal fin in boxfishes differ from that of other fishes? How are the fluid dynamics of steering fundamentally different from propulsion, and how are the boxfishes taking advantage of this? This is an issue of framing. The authors give a very slight nod to these ideas (lines 89-91 and lines 326-328), while devoting a huge portion of the text to discussing stability vs maneuverability. The authors could have put their research into the context of the swimming literature over the last century instead of focusing on the last decade, to increase the broad interest of this study.

→ **We fully agree with the referee that ostraciiform locomotion, as classically categorised by Breder (1926), is mainly median/paired fin locomotion instead of body/caudal fin locomotion driven. However, Breder (1926) does mention the following on page 171: "Since the Ostraciidae are incased in hard and inflexible tests it is obviously impossible for them to pass waves posteriorly as do the more flexible fishes. The tail protrudes from an opening in the test and is supported on a peduncle too short to be thrown into waves and is therefore insufficient to give any efficacious movement of the anguilliform type. In these fishes the locomotor emphasis is placed on the pectoral fins and other parts while the tail is used primarily as a rudder. However, at times when more than ordinary speed is required, the tail is given the only motion possible, a lashing from side to side."**

→ **Hereafter Breder (1926) elaborates on the use of the body and caudal fin in propulsion and how models show that this could be possible; which is further quantified by Blake (1981). But he does state, as shown above, that the other fins apart from the caudal fin are mainly used for propulsion during non-fast swimming. This is indeed demonstrated by Gordon et al. (2000) and Walker (2000), but Blake (1977) already touches upon this: "Breder (1926) showed that a rigid fin oscillated symmetrically about its base will produce a propulsive thrust. However, his model of ostraciiform locomotion is only adequate for the caudal fin during fast forward progression. All the other fins show wave-forms (often as much as three-quarters of a complete wavelength in the case of the dorsal and anal fins) in addition to a simple side to side movement." In this context, the caudal fin of boxfishes**

does indeed represent an extreme on this spectrum in which they use it as a rudder, which is what we find.

→ We added more information on the state of the caudal fin during forward swimming and yaw manoeuvres in the discussion section (see above) to emphasise that the usage of the caudal fin in living animals does indeed matches our model findings. The focus of current paper is yaw turning and not necessarily propulsion, nevertheless we added a few sentences on this in the discussion: “The seemingly active lateral movements of the caudal fin during fast swimming as observed by Gordon [13] and Blake [19] indicates body/caudal fin propulsion as originally termed by Breder [28] as ostraciiform locomotion [68]. However, apart from fast burst-and-coast swimming, propulsion is median/paired fin driven, engaging the caudal fin as rudder [13,19,20,28]. This role of the caudal fin in swimming is in accordance with the large yaw torques generated due to the presence of the caudal fin as measured in the current study.”

References:

- Blake RW. 1977 On ostraciiform locomotion. *J. Mar. Biol. Assoc. UK.* 57, 1047–1055. (doi:10.1017/S0025315400026114)
- Blake RW. 1981 Mechanics of ostraciiform propulsion. *Can. J. Zool.* 59, 1067–1071. (doi:10.1139/z81-148)
- Breder CM. 1926 The locomotion of fishes. *Zoologica* 4, 159–297.
- Gordon MS, Hove JR, Webb, PW, Weihs D. 2000 Boxfishes as unusually well-controlled autonomous underwater vehicles. *Physiol. Biochem. Zool.* 73, 663–671. (doi:10.1086/318098)
- Walker JA. 2000 Does a rigid body limit maneuverability? *J. Exp. Biol.* 203, 3391–3396.

During the previous round of reviews, all three reviewers took issue with the authors' handling of the "stability vs maneuverability" issue. They point out that (1) the authors misrepresent the previous work in this area and (2) the authors' present study is limited in its ability to draw conclusions about stability and maneuverability, given the lack of data on dynamic control of maneuvering in boxfishes. I feel that the authors have made good progress towards addressing both of these issues in their revision, but I have a few suggestions to further improve the manuscript. As mentioned above, the authors could present the study in the broader context of the role of the caudal fin in fish swimming (especially steering vs. propulsion) and how boxfishes are unique. This would allow the authors to put their results into a much broader context of the fish swimming literature. As the other reviewers point out, little can be said about stability and maneuverability from this dataset, but the use of the caudal fin as a rudder is clearly demonstrated by this study.

→ We thank the referee for the positive and encouraging words on our improvements made to the manuscript compared to the previous version. We have added additional information on the broader context of the role of the caudal fin in fish swimming to the manuscript and the unique position of boxfishes by mainly engaging the caudal fin as a rudder instead of using it for propulsion as described above.

→ However, we feel that with the knowledge on the stability of the carapace (without fin), which does generate stabilising vortices but is net unstable, in combination with engaging the caudal fin as a rudder to modulate this instability, we can address the stability and manoeuvrability of the yellow boxfish. We hypothesise that, due to the stabilising caudal fin and inherently net unstable carapace, the yellow boxfish can increase its manoeuvrability and swim in a stable manner. This perspective is new since previous studies on the stability of boxfish only considered the role of the carapace without fins whilst now we can build further on this by quantifying the role of – as in our current case – the caudal fin. In addition, based on the suggestion of referee 2 (see below), we changed the word ‘manoeuvrability’ into ‘yaw stability’ in the title of the manuscript.

In terms of addressing the previous work of Bartol and colleagues, my primary issue is with the repeated assertion that the keel-induced vortices of the boxfish carapace have a "negligible effect" on yaw. Bartol observed these vortices and found them to be important in self-correcting, especially for pitch but also for yaw. He did not assert that the entire carapace had a net stabilizing effect. Therefore, the "negligible effect" of these vortices documented by Van Wassenbergh et al. (2015) is not a direct rejection of Bartol's findings. Also, although Van Wassenbergh characterizes the effect of the vortices as "negligible" in his 2015 study, his graphs of yaw and pitch moments do show a stabilizing effect of the keel vortices. While the stabilizing moments were certainly less than the destabilizing moments, they were nowhere close to zero. In fact, Van Wassenbergh et al. (2015) acknowledges that they "could confirm the previously predicted stabilizing effect on the lateral carapace walls at the posterior end of the fish." However, no statistical analysis or evaluation was conducted to support the assertion that the stabilizing moments can be accurately described as "negligible" relative to the destabilizing moments. Furthermore, as the authors point out, none of the previous studies have included the large and distinctive boxfish scales, which likely have a large impact on flow separation and could drastically change modeling results. Therefore, the rejection of self-correcting keel vortices as "negligible" seems premature without additional work. It is outside the scope of the present study to address this, but I strongly encourage the authors to consider presenting a more balanced perspective on the literature. It is especially important for the authors to be cautious here, because they did not make direct observations of vortices, nor did they study how these vortices might interact with the caudal fin, despite mentioning this in the introduction.

- **We fully agree that it can be discussed whether 'negligible' is an appropriate term. Therefore, we replaced 'negligible' in our manuscript by "relatively small".**
 - **The statement that Bartol *et al.* 'did not assert that the entire carapace had a net stabilizing effect' is not correct: all his flow-tank measurements of torque are for entire carapaces, and these all are reported as being stabilising (i.e. net stabilising for the entire carapax). For an independent assessment of this topic, we would like to refer to the article by Farina & Summer (2015) referenced in the our manuscript.**
 - **We agree that small-scale surface structures due to the presence of the surface scutes (i.e. see Yang *et al.* (2015) for more information as these are not fully comparable to scales) have not been included in the present study. However, the insensitivity of the model to differences in surface roughness has been tested and reported in Van Wassenbergh *et al.* (2015): "As increasing the roughness of the boxfish surface (from 0 to 2 mm) or increasing turbulence intensity of the incoming flow (from 5 to 50%) both further increased the calculated destabilizing moments by a few percentages, these parameters have no influence on the conclusions of our study." This is mentioned in the manuscript and we now also included this reference: "The effect of adding a natural surface roughness remains untested, but (uniform) modifications of surface roughness constants in computational fluid dynamics models (see below for details on this method), suggest a negligible effect on overall yaw torque [11]."**
 - **Quantifying the effects of having keels vs. no keels is indeed outside of the current scope.**
- Reference:

- Farina SC, Summers AP. 2015 Biomechanics: Boxed up and ready to go. *Nature* 517, 274–275. (doi:275.10.1038/517274a)
- Van Wassenbergh S, van Manen K, Marcroft TA, Alfaro ME, Stamhuis EJ. 2015 Boxfish swimming paradox resolved: forces by the flow of water around the body promote manoeuvrability. *J. R. Soc. Interface* 12, 20141146. (doi:10.1098/rsif.2014.1146)
- Yang W, Naleway SE, Porter MM, Meyers MA, McKittrick J. 2015 The armored carapace of the boxfish. *Acta Biomater.* 23, 1–10. (doi:10.1016/j.actbio.2015.05.024)

In terms of technical details, I believe that the authors have adequately addressed previous issues with this paper. My only remaining technical concern is that I would like to see more information on the properties of the PMMA tail. Considering that this paper relies only on models, the description of "this material is relatively stiff but still bends a bit at larger forces" is not adequate. Did the material bend at the forces experienced at the highest flow speeds? Was any deformation observed during the experiment? Were fluid-solid interactions included in the computational model to address bending of the caudal fin? My general understanding is that fishes use muscles to control the shape of the tail and resist passive bending due to incoming flow, and therefore an inflexible material may have been more appropriate, unless the authors have previously observed passive bending of the caudal fin in boxfishes. Again, this is where observations of live animals are necessary and useful.

→ **We are happy to hear that the referee is positive regarding our improvements made compared to the previous version of the manuscript. We agree with the referee that our previous description of the caudal fin material was rather crude. We have added the Young's modulus of the PMMA to the manuscript, which is ~3 GPa. The Young's modulus is a mechanical property that measures the stiffness of a solid material and is independent from the material's thickness. Together with the thickness of the PMMA used, i.e. 1 mm, this should provide the reader with adequate information to compare our data with other studies. As mentioned in our response to the associate editor, we argue that the questions we want to answer with our study do not necessitate an analysis that includes all details of a natural fin. Instead we explore the functional implications of caudal fin position and size on yaw, for which some simplifying assumptions (which we hope to have described in sufficient detail) do not invalidate our study. Finer control of the shape of the fin, or passive mechanical effects are indeed very interesting, but we hope that the reviewer understands that analysing these effects are beyond the scope of the current study.**

Minor comments:

The authors do a relatively good job of using "boxfishes" when generalizing about multiple species, but this should be checked throughout (see "boxfish" on lines 21, 29, 64, 70, 77, etc).

→ **We agree with the referee and changed this accordingly, if deemed necessary, in the revised manuscript, as highlighted in orange.**

When discussing the choice of which flow speeds to use, the authors focus on reported fish swimming speeds. However, isn't the focus of this study on destabilizing incoming flows that might be experienced in a complex reef environment? How do the flow speeds reflect those kinds of interactions?

→ **The referee is indeed correct that the movement of water relative to the animal matters, and this can result from both swimming in standstill water as from holding position with incoming flows (as two extremes of a continuum). However, in previous studies the choice of flow speeds was based on the swimming speeds that boxfishes could achieve. We followed the referee's advice, and searched for literature on flow speeds in coral reefs. According to Hench & Rosman (2013) and Comeau *et al.* (2014), the mean flow speed on a reef in French Polynesia (overlapping with the yellow boxfish distribution according to Myers (1999)) was 0.15 m s^{-1} and 0.1 m s^{-1} respectively. In addition, water flow speeds on a site in the northern Great Barrier Reef in Australia (also overlapping with the yellow boxfish distribution) varied between ~ 0.1 to $\sim 0.4 \text{ m s}^{-1}$ (Heatwole & Fulton, 2013; Johansen, 2014). Unfortunately, to our best knowledge, no literature is available on the water flow speeds in relation to (yellow) boxfish swimming behaviour and (micro)habitats**

specifically. Flow speed measurements on Caribbean reefs (i.e. not overlapping with the distribution of yellow boxfish) show that these may vary within the range of $\sim 0.1\text{--}0.5\text{ m s}^{-1}$ but may also be lower depending on the microhabitat (Sebens *et al.*, 1997; Finelli *et al.*, 2009). Nevertheless, since overall the observed relationships are similar for slow water flow speeds (0.1 m s^{-1}) up to fast flow speeds (0.5 m s^{-1}), the choice of flow speed does not influence our conclusions. We added the following sentence on flow speeds on coral reefs to the manuscript: "These speeds also correspond roughly to flow speeds measured in coral reef habitats [41–44] which overlap with the yellow boxfish distribution according to Myers [25]."

References:

- Comeau S, Edmunds PJ, Lantz CA, Carpenter RC. 2014 Water flow modulates the response of coral reef communities to ocean acidification. *Sci. Rep.* 4, 6681. (doi:10.1038/srep06681)
- Finelli CM, Clarke RD, Robinson HE, Buskey EJ. 2009 Water flow controls distribution and feeding behavior of two co-occurring coral reef fishes: I. Field measurements. *Coral Reefs* 28, 461–473. (doi:10.1007/s00338-009-0481-0)
- Heatwole SJ, Fulton CJ. 2013 Behavioural flexibility in reef fishes responding to a rapidly changing wave environment. *Mar. Biol.* 160, 677–689. (doi:10.1007/s00227-012-2123-2)
- Hench JL, Rosman JH. 2013 Observations of spatial flow patterns at the coral colony scale on a shallow reef flat. *J. Geophys. Res. Oceans.* 118, 1142–1156. (doi:10.1002/jgrc.20105)
- Johansen JL. 2014 Quantifying Water Flow within Aquatic Ecosystems Using Load Cell Sensors: A Profile of Currents Experienced by Coral Reef Organisms around Lizard Island, Great Barrier Reef, Australia. *PLoS ONE* 9, e83240. (doi:10.1371/journal.pone.0083240)
- Myers RF. 1999 *Micronesian reef fishes: A comprehensive guide to the coral reef fishes of Micronesia* (3rd revised and expanded edition). Barrigada, Guam: Coral Graphics. 522 pp.
- Sebens KP, Witting J, Helmuth B. 1997 Effects of water flow and branch spacing on particle capture by the reef coral *Madracis mirabilis* (Duchassaing and Michelotti). *J. Exp. Mar. Biol. Ecol.* 211(1), 1–28. (doi:10.1016/S0022-0981(96)02636-6)

Lines 40-41 - The uses of the "/" symbol to indicate "and/or" are not appropriate. The terms "adapted," "specialised," "swimming modes," and "functions" have distinct definitions and cannot be used in an "and/or" context. The authors should choose the most accurate term in both cases or elaborate on the intended meaning. I encourage the authors to avoid the "and/or" construct throughout the rest of the paper.

➔ We agree with the referee and changed this accordingly in the revised manuscript, as highlighted in orange. However, we did not change the terms "pectoral/anal-fin-dominant swimming" and "anal/dorsal-fin-dominant swimming" in lines 60-61 as this is commonly used in literature (e.g. Hove *et al.*, 2001).

References:

- Hove JR, O'Bryan LM, Gordon MS, Webb PW, Weihs D. 2001 Boxfishes (Teleostei: Ostraciidae) as a model system for fishes swimming with many fins: Kinematics. *J. Exp. Biol.* 204, 1459–1471.

Line 51 - Should read "ostraciid"

➔ We fully agree with the referee and changed this word accordingly.

Lines 97-110 - It would be helpful if the authors would outline their many experiments explicitly in this paragraph, to provide a road map for the readers for this complex experimental design. For example, there is no mention of the tailless experiments in this paragraph.

→ **We agree with the referee and changed the paragraph accordingly in the revised manuscript, as highlighted in orange.**

Line 436 - The authors should also include STL or OBJ files for their 3D models, which are necessary for replication.

→ **We agree with the referee and have included the STL file and IGES files used for the construction of the physical model and used in the computation fluid dynamics modelling.**

Reviewer: 2

Comments to the Author(s)

Overall Comments:

This manuscript investigates the effect of body angle and tail angle on the generation of yawing torques in a physical model of boxfishes. The major findings are as follows: (1) Yaw about a rigid body is generally destabilizing when the body has a non-zero angle of attack relative to incoming flow. (2) When a flexible caudal fin (though not one tuned to the properties of a biological fin) is added to the rigid model, the configuration becomes self stabilizing at increasing angles of attack. (3) Including a tail angle can create self-stabilizing yaw torques (negative yaw torques at positive body angles of attack).

While the modeling done for this study is mostly sound, I have specific concerns about the position of the sting and its effects on yawing moments; the lack of information about the flexural stiffness of the caudal fin model; and the use of the tail-less model as a kind of null for the hypothesis of self-stability. In addition, I question how informative this research is over previous studies. We already know that fins induce yawing moments. It does not seem like a notable advance to quantify those moments in a model that may not match biological parameters observed in fishes (like tail stiffness).

- **We regret that the reviewer did not perceive our results as a notable advance. Because our work has been preceded by several publications in important journals aiming exclusively at quantifying the torque on boxfish carapaces, we feel that our study is important to put these measurements into the broader perspective of how much, and under which conditions (position, closed versus open) the caudal fin can alter the animal's net torque for yaw.**
- **As mentioned earlier, we acknowledge that not all biological complexity could be accounted for, but would like to note that our model does match several important biological parameters: it has a realistic carapace shape based on 3D scans of specimens and fin position, contour shape, and surface area have been matched to photographs of a living specimen.**
- **To add information on the flexural stiffness of our model, we now included the Young's modulus of the material used for our fins.**

Specific Comments:

Title: The title seems misleading here. This paper, to me, seems to be looking at modulation of passive stability in yaw, not maneuverability. In other words, I think of maneuverability as what the living fish will actually do to generate torques (actual behavior), not what it is hypothetically capable of doing.

→ **On the one hand, stability and manoeuvrability are commonly used as antonyms in the literature of fish swimming (e.g. Weihs, 2002), so in this sense we could choose both manoeuvrability and stability. Linguistically, manoeuvrability means ‘the quality of being able to be manoeuvred easily while in motion’ and thus points to a capability rather than an actual behaviour. In this sense, we also see no problem with the current title. On the other hand, we think we understand the feeling of the reviewer that we have not really dealt with the actual execution of the manoeuvre. Still, considering all other steering elements (e.g. action of other fins) to remain unchanged, our ‘manoeuvrability’ will be proportionate to this ‘manoeuvring performance’. Yet, since we do not have a preference ourselves, we are happy to follow the referee’s suggestion and changed ‘manoeuvrability’ into ‘yaw stability’.**

Reference:

- **Weihs D. 2002 Stability versus maneuverability in aquatic locomotion. *Integr. Comp. Biol.* 42, 127–134. (doi:10.1093/icb/42.1.127)**

Ln 54-56: The first half of this paragraph can be considered relevant context in that the armor plating of boxfishes inevitably will contribute to the range of motion of the body – but the discussion of ostracitoxin seems irrelevant in this context.

→ **We agree with the referee and removed these two sentences in the revised manuscript here. We integrated them into the discussion section, as indicated in orange, where we do discuss the role of ostracitoxin in the broader context of boxfish ecology.**

Ln 67-69: Suggest removal of the “the intended change...in a controlled manner”.

→ **We followed the suggestion of the referee and removed this in the revised manuscript.**

Ln 97-110: These should be in the past tense.

→ **We agree with the referee and changed this accordingly in the revised manuscript, as highlighted in orange. However, we only changed the tense where we describe the outline of the materials & methods. E.g. we did not change the tense in the last sentence of this paragraph since we describe our hypothesis here.**

Ln 137-140: Why not place the sting at the center of mass, if its position in the fish specimens was known? This seems an avoidable source of error. I am concerned that even small differences in the position of the sting may have consequences for the location of the “switch points” discussed later on.

→ **The position of the centre of mass was not known in the fish specimens nor in the model. The centre of mass is also not constant due to potential swim bladder expansion and body shape (e.g. the angle of the tail). The accuracy of our estimate of centre of volume as an approximation of the centre of mass is discussed in detail in the response-to-reviews letter on the previous round of reviews.**

Ln 157: Does the caudal fin scale isometrically in the fish? Absence of studies of scaling does not mean evidence of isometry.

→ We assumed that the caudal fin scales isometrically but could not check this since the living specimens in the laboratory were of similar size. Nevertheless, as we mentioned in our response letter to the previous round of reviews, absolute size difference between the photographed living fish and our models was small, and therefore allometry would result in a small error during scaling from the living specimens to the model when falsely assuming isometry. We added to the manuscript that the isometric scaling was by a few percent.

Ln 161-164: The flexural stiffness of the PMMA, or at least the Young's modulus, could be compared to the measured values for actual fishes. (e.g. Tangorra et al. 2010)

→ We have added the Young's modulus of the PMMA (which is independent from material thickness) to the manuscript, which is ~3 GPa, as mentioned in response to referee 1. In addition, the thickness of the caudal fin was already provided in the manuscript, i.e. 1 mm. These two values should enable comparison between studies.

Reference:

- Tangorra JL, Lauder GV, Hunter IW, Mittal R, Madden PGA, Bozkurttas M. 2010 The effect of fin ray flexural rigidity on the propulsive forces generated by a biorobotic fish pectoral fin. *J. Exp. Biol.* 213, 4043–4054. (doi:10.1242/jeb.048017)

Ln 319: But others have found self-stabilizing flows over the carapace of boxfishes when including the caudal fin. It seems like a straw-man argument to draw “adaptive” conclusions about the relative stability of the boxfish carapace when it evolved in the context of having a caudal fin.

→ Self-stabilising flows have indeed been measured previously by Bartol *et al.* (2002, 2003, 2005), but only for models without caudal fin. However, these measurements appeared not reproducible by Van Wassenbergh *et al.* (2015), both using CFD and in a flow tank with physical models. These physical and CFD models neither had a caudal fin but the net effect was non-stabilising.

→ As mentioned in our previous response-to-referees letter: “We disagree with the reviewer on this point. Forces and torques measured by Bartol *et al.* were always without tail: Bartol *et al.* (2003): “tail of the model was removed and replaced with a 10 cm rod (also painted black and 1.0 cm in diameter, which was similar in dimension to the caudal peduncle). The smooth trunkfish model used in DPIV experiments was also used in force measurement experiments.” The same methods are used in their article from 2002 (*Integr. Comp. Biol.* 42) and the one from 2005 that compared different species. Their article from 2008, the one Webb and Weihs (2015) refer to in this specific sentence indeed includes experiments on living fish (flow visualization using PIV), but does not include force or torque measurements. So in contrast to what the referee suggests, there have not been force and torque measurements from Bartol and colleagues on models with a tail.”

References:

- Bartol IK, Gordon MS, Gharib M, Hove JR, Webb PW, Weihs D. 2002 Flow patterns around the carapaces of rigid-bodied, multi-propulsor boxfishes (Teleostei: Ostraciidae). *Integr. Comp. Biol.* 42, 971–980. (doi:10.1093/icb/42.5.971)
- Bartol IK, Gharib M, Weihs D, Webb PW, Hove JR, Gordon MS. 2003 Hydrodynamic stability of swimming in ostraciid fishes: role of the carapace in the smooth trunkfish *Lactophrys triqueter* (Teleostei: Ostraciidae). *J. Exp. Biol.* 206, 725–744. (doi:10.1242/jeb.00137)

- Bartol IK, Gharib M, Webb PW, Weihs D, Gordon MS. 2005 Body-induced vortical flows: a common mechanism for self-corrective trimming control in boxfishes. *J. Exp. Biol.* 208, 327–344. (doi:10.1242/jeb.01356)
- Van Wassenbergh S, van Manen K, Marcroft TA, Alfaro ME, Stamhuis EJ. 2015 Boxfish swimming paradox resolved: forces by the flow of water around the body promote manoeuvrability. *J. R. Soc. Interface* 12, 20141146. (doi:10.1098/rsif.2014.1146)

Appendix B

We thank the reviewers and editor for their valuable suggestions on how to improve this article. We appreciate their time and effort for making these reviews. We hope that our revision of the manuscript and clarifications in this response-to-referees letter make it suitable for publication in the Royal Society Open Science.

In the text below, the original review reports are given, with our replies printed in bold after an indented arrow. All changes to the text are printed in orange in the uploaded manuscript file. In addition, we also uploaded a 'clean' version.

Associate Editor Comments to Author (Professor Brooke Flammang):

Both reviewers have seen that the authors have made some efforts to revise their paper. However, of a more major concern is that there are some important issues with regards to arguments laying foundation for the major findings of this paper that have been rebutted in the response instead of clarified within the manuscript itself. I encourage the authors to consider that whether they think the reviewers are more correct in their interpretation or not, it is the role of the author is to provide a clear explanation of the science being discussed. If several reviewers have brought up similar issues disagreeing with the authors' point of view, it seems that the explanation provided in the manuscript is perhaps not as clear as it could be.

Subject Editor comments:

Thanks for your revisions. It is very important to lay out the work and arguments of previous research clearly, and our reviewers and AE think that you have not yet brought the manuscript to that point. Please provide your best explanation in your revision. We will be unable to send it out for another round otherwise. Thanks again.

→ We are sorry to hear that some points have not been completely resolved in our previous revision. Below we wrote how we dealt with the remaining remarks.

Reviewer comments to Author:

Reviewer: 1

Comments to the Author(s)

This article about the hydrodynamics of the caudal fin of the yellow boxfish is an interesting study that offers some insights into the utility of the large caudal fin of boxfishes, in coordination with the rigid carapace. Using models, the authors make a compelling case that the caudal fin could act as a stabilizer in response to incoming flow.

The authors did an adequate job with this revision. They added some crucial information in support of their study. I still have a few concerns, described below. However, the authors have made their decisions in terms of how to frame the study and how to address the literature. It is now up to the editors to decide if the article is suitable for their audience and meets their standards of literature

review. My assessment is that the article is scientifically sound and suitable for publication, in addition to providing valuable information about boxfish swimming hydrodynamics.

A major improvement to the manuscript is the addition of information about the position of the caudal fin in live boxfishes from the literature. The authors provide details of Blake's observations of boxfish caudal fins (i.e., the behaviors during which the fin is open or closed and the degree to which the fin is angled). This context provides support for the realism of this model. Previously, the article was written as though there was no documentation of when and how an open or closed caudal fin was used by boxfishes, which made it difficult to evaluate the utility of this model. I encourage the authors to incorporate a summary of Blake's observations into the introduction, because they are crucial to interpreting the experiments. The current statement (lines 92-95) is too vague.

→ **We agree with the referee and have incorporated a summary of the observations of Blake (1977) into the introduction of the revised manuscript as highlighted in orange.**

References:

- **Blake RW. 1977 On ostraciiform locomotion. *J. Mar. Biol. Assoc. UK.* 57, 1047–1055. (doi:10.1017/S0025315400026114)**

I am glad that the authors have provided a Young's modulus for the PMMA material. However, the authors did not address my questions from the previous review: "Did the material bend at the forces experienced at the highest flow speeds? Was any deformation observed during the experiment?" Since the authors state that the material "is relatively stiff but still bends a bit at larger forces," I would like to know if any bending was observed. Considerable bending could have a significant impact on interpretation of the results. If the authors observed minimal to no bending, this should be reported to address this concern.

→ **We apologise to have failed answering to the reviewer's questions on bending of the caudal fin. Bending of the caudal fin was not quantified, but bending was minimal (estimated < 3 mm deflection of the caudal fin tip) and only occurred at the highest flow speed when the caudal fin was oriented relatively perpendicularly to the incoming flow. We added the previous sentence to the Material & Methods section of the manuscript as highlighted in orange for clarification on the observed bending of the caudal fin.**

→ **We also noticed to have failed to answer another question of the reviewer on this topic in the previous round, namely: "Were fluid-solid interactions included in the computational model to address bending of the caudal fin?". No fluid-solid interaction was incorporated in the CFD, hence the caudal fin was completely rigid and flat (i.e. could not bend).**

→ **Taking into account the above, we deem the observed minimal bending to not affect the conclusions of our study. In addition, the Young's modulus of the PMMA together with the PMMA thickness, i.e. 1 mm, this should provide the reader with adequate information to compare our data with other studies and enable repetition of the measurements.**

I still feel that the "stability vs maneuverability" framework does not serve this paper well. I feel that the authors have demonstrated the use of the caudal fin as a rudder for steering, as the authors repeatedly state. However, the utility of the caudal fin in dynamic control of course stability is still unknown and is, as the authors state, outside the scope of this study. However, the authors have made their case for this framing.

It seems unnecessary and inappropriate to repeatedly reject the findings of previous studies of carapace hydrodynamics, as opposed to building upon and updating previous understandings. As many previous referees have pointed out, the authors could provide a more balanced and nuanced discussion rather than outright rejection of the work by Bartol and colleagues.

I find no evidence in any of the Bartol articles or in Farina and Summers (2015) that Bartol and colleagues asserted that the entire carapace had a net stabilizing effect, despite the authors repeatedly making this claim in response to reviewers. As Farina and Summers point out, Bartol and colleagues focus on their discovery of stabilizing vortices associated with the keels, and they provide ample evidence of this. While Van Wassenbergh (2015) provides strong evidence of a net destabilizing effect of carapace hydrodynamics as a whole, which is a major contribution to our understanding of boxfish biology, this does not refute the presence of the keel vortices or their potential to provide stabilization in some contexts (either in biomimetic design or in the biology of the animals). More research is clearly needed. As Farina and Summers point out (and as summarized in their figure), Van Wassenbergh's study provides strong evidence against the self-stabilizing nature of the entire carapace, but that does not mean that the entirety of Bartol and colleagues previous work on keel vortices is invalidated.

- **In the manuscript we wrote that “Previous studies hypothesised that the flow over the carapace of boxfishes induces course-stabilising and self-corrective trimming vortices during pitching (i.e. rotation about the lateral axis) and yawing (i.e. rotation about the dorso-ventral axis) [14,15,22,23].” We would like to note that torque measurements showing that the entire carapace had a net stabilising effect for pitching can be found in:**
- **Bartol *et al.* (2002) on page 979 Fig. 6A. These data are explained in the results section on page 976 and interpreted in combination with the measured vortices in the discussion on page 979 as follows: “Consequently, suction derived from the presence of a vortex above or below the ventral keels should act largely upward and posterior to the center of mass at positive angles of attack (which also occurs in delta wings) and downward and posterior to the center of mass at negative angles of attack. Based on pitching moments recorded in force balance experiments, where nose-down pitching moments occurred and became progressively stronger as angles of attack became more positive and nose-up pitching moments occurred and became progressively stronger as angles of attack became more negative, this is exactly what happens. Therefore, the ventral keels**

and in some species the dorsal keels are effectively generating self-correcting forces for pitching motions; the degree of self-correction is proportional to the degree to which the fish is perturbed from a horizontal swimming trajectory.”

- Bartol *et al.* (2003) on page 740 Fig. 13D
- Bartol *et al.* (2005) on page 340 Fig. 8C

→ Similar effects for yaw torque, although not measured by Bartol and colleagues, were hypothesised in these publications based on similarity of the pressures and shapes of the vortices when yawing as measured during pitching. Hence we explicitly wrote in our manuscript that these effects are ‘hypothesised’. See below for an example highlighting the measurements and conclusions regarding yaw:

- Bartol *et al.* (2003) wrote in the abstract on page 725: “Furthermore, nose-down and nose-up pitching moments about the center of mass were detected at positive and negative pitching angles of attack, respectively. The three complementary experimental approaches all indicate that the carapace of the smooth trunkfish effectively generates self-correcting forces for pitching and yawing motions – a characteristic that is advantageous for the highly variable velocity fields experienced by trunkfish in their complex aquatic environment.”

→ We hope that by referring the reviewer to the precise position of these data from the literature, he/she will agree that our interpretation of the literature is not incorrect. This view also corresponds to the text by Farina and Summers (2015) writing that “A series of previous studies had suggested that the boxfish carapace is self-stabilizing”, which in our opinion cannot refer to local contributions of forces by vortices but is referring to the overall carapace. However, as mentioned in our previous response-to-referees letter, we fully agree that our manuscript is not addressing the role of keel vortices, and therefore it certainly does not invalidate these parts of the work by Bartol and colleagues. We added a new sentence highlighting this in the manuscript (lines 405-407): “The previously discovered vortices that cause self-corrective forces on the carapace [14,15,22,23] will contribute to reduce such recoil motions.”

References:

- Bartol IK, Gordon MS, Gharib M, Hove JR, Webb PW, Weihs D. 2002 Flow patterns around the carapaces of rigid-bodied, multi-propulsor boxfishes (Teleostei: Ostraciidae). *Integr. Comp. Biol.* 42, 971–980. (doi:10.1093/icb/42.5.971)
- Bartol IK, Gharib M, Weihs D, Webb PW, Hove JR, Gordon MS. 2003 Hydrodynamic stability of swimming in ostraciid fishes: role of the carapace in the smooth trunkfish *Lactophrys triqueter* (Teleostei: Ostraciidae). *J. Exp. Biol.* 206, 725–744. (doi:10.1242/jeb.00137)
- Bartol IK, Gharib M, Webb PW, Weihs D, Gordon MS. 2005 Body-induced vortical flows: a common mechanism for self-corrective trimming control in boxfishes. *J. Exp. Biol.* 208, 327–344. (doi:10.1242/jeb.01356)
- Farina SC, Summers AP. 2015 Biomechanics: Boxed up and ready to go. *Nature* 517, 274–275. (doi:275.10.1038/517274a)

Also, as a side note, while I agree that Farina and Summers provide a balanced perspective, it may not be appropriate to cite a News and Views article in this context. On line 395, it is cited as if it were a research article, which seems a bit misleading.

→ **We agree with the referee and have adjusted the manuscript accordingly by highlighting the type of paper: “However, the entire carapace acts destabilising for yaw (Fig. 5) and pitch [11], as reviewed in the News and Views article of Farina & Summers [45]. Hence performing turning manoeuvres will require less force from the fins, or will result in faster turns for a given force input from the fins.”**

As for data accessibility, is there something that you can do to reduce the file size for download from DataDryad? 40 GB is a lot. Perhaps files could be grouped and stored as separate DataDryad entries, so that not all files need to be downloaded at once.

→ **We agree with the referee that this is inconvenient and were uninformed of this issue when uploading our data. We have enquired DataDryad about this issue to which they replied with the following: “It is true that the temporary review URL is a direct download link that downloads the entire dataset at once. However, when the dataset is published it will be possible to download the files individually. See for example <https://doi.org/10.5061/dryad.pg4f4qrjq> -- if you expand the file list in the "Data files" box (under the date), you can download individual files from there.”**

→ **Hence the supplementary files will be available for downloading individually when the dataset is published. Should the reviewer like to access and/or download the files individually now, he/she can access the files through the following OneDrive link: <https://1drv.ms/f/s!ArePxNJHFji7mmMdITTUIUoLQ3RI>.**

Reviewer: 2

Comments to the Author(s)

While previous reviewers have provided a wealth of feedback to improve this manuscript, the authors have by and large chosen to rebut this feedback, and reassert some of their more problematic claims. As a result, I am hesitant to recommend this manuscript for publication.

A previous reviewer suggested reframing in the context of the historical literature (ostraciiform locomotion writ large) that would alleviate much of this issue. In short, my concerns with this manuscript are less to do with the experiments, and more to do with the interpretation of the data in the context of the literature.

The authors maintain that both the reviewers on this manuscript, and those of the previous (2015) manuscript have incorrect assertions about the literature regarding stability and maneuverability. This is deeply troubling: if that many professionals in swimming biomechanics (the reviewers) disagree with the authors' interpretation, then the authors need to do a better job justifying their interpretation, beyond citing their own paper. The authors suggest that Farina and Summers provide an "independent assessment" of this topic, but the article they reference is merely a summary of their article, and not an independent assessment. This manuscript would be made much stronger by a more balanced perspective on the literature, but the authors do not appear willing to take this approach.

- **In our opinion it is logical that reviewers are critical and sceptic when new data are presented that not entirely in line with previous experimental measurements. Concerning our previous manuscript from 2015 in J R Soc Interface, please note that we successfully addressed all criticisms in our revision and response letter. Concerning the remaining 'incorrect assertions about the literature' from the current reviews, we refer to our response to the first reviewer, in which we assume that data on torques of entire carapaces in the work by Bartol and colleagues were overlooked by the reviewer. We agree that the reference to Farina and Summers was not entirely appropriate since this is a News and Views report and have adjusted the manuscript accordingly as highlighted in orange. We feel that we have made adjustments in our previous revision to provide a more balanced perspective, e.g.: concerning the previously termed 'negligible' effect of keel vortices that was consistently rephrased; by replacing 'manoeuvrability' with 'yaw stability' in the title; by adding information about the position and state of the caudal fin in live boxfishes. In addition, as mentioned in our response to the first reviewer, we added a new sentence highlighting that our manuscript does not invalidate the roles of the keel vortices (lines 405-407): "The previously discovered vortices that cause self-corrective forces on the carapace [14,15,22,23] will contribute to reduce such recoil motions."**
- **But, as our data are the first yaw torque measurements on boxfish with caudal fin, we hope that the reviewer agrees that our manuscript is centered around the currently presented data and its interpretation, and that information from previous literature on this topic is scarce.**

References:

- **Farina SC, Summers AP. 2015 Biomechanics: Boxed up and ready to go. *Nature* 517, 274–275. (doi:275.10.1038/517274a)**
- **Van Wassenbergh S, van Manen K, Marcroft TA, Alfaro ME, Stamhuis EJ. 2015 Boxfish swimming paradox resolved: forces by the flow of water around the body promote manoeuvrability. *J. R. Soc. Interface* 12, 20141146. (doi:10.1098/rsif.2014.1146)**

Appendix C

We thank the reviewers and editor for their valuable suggestions on how to improve this article. We appreciate their time and effort for making these reviews. We hope that our revision of the manuscript and clarifications in this response-to-referees letter make it suitable for publication in the Royal Society Open Science.

In the text below, the original review reports are given, with our replies printed in bold after an indented arrow. All changes to the text are printed in orange in the uploaded manuscript file. In addition, we also uploaded a 'clean' version.

On behalf of the Editor, I am pleased to inform you that your Manuscript RSOS-200129 entitled "Modulation of yaw stability with an unstable rigid body and a stabilising caudal fin in the yellow boxfish (*Ostracion cubicus*)" has been accepted for publication in Royal Society Open Science subject to minor revision in accordance with the referee suggestions. Please find the referees' comments at the end of this email.

The reviewers and Subject Editor have recommended publication, but also suggest some minor revisions to your manuscript. Therefore, I invite you to respond to the comments and revise your manuscript.

Subject Editor Comments to Author (Professor Kevin Padian):

Comments to the Author:

Dear authors, we will be happy to accept your manuscript but I would like you to consider and respond to the comments of the single reviewer who still has reservations when you resubmit your manuscript in final form. Thanks very much.

→ **We are happy to hear that our manuscript is accepted for publication in Royal Society Open Science. Below we wrote how we dealt with the remaining remarks of the reviewers. We also rewrote some sentences to increase readability.**

Reviewer comments to Author:

Reviewer: 1

Comments to the Author(s)

The authors have adequately responded to all of my concerns, and I am satisfied with this revision. I still have a minor concern, but the authors can decide whether they want to address it, and there is no need for me to see another revision.

In my last set of comments, one of my concerns was that the authors still characterized the work by Bartol and colleagues to have concluded that the shape of the carapace had a net-stabilizing effect. However, there is confusion about "net stabilizing" vs "self-correcting." Bartol et al certainly repeatedly claimed that the vortices were self-correcting and self-stabilizing, and Van Wassenbergh shows that the carapace as a whole had net-destabilizing tendencies, a subtle distinction that was summarized briefly in the Farina and Summers review (in particular, see their figure). However, I'm still not convinced that their goal was ever to show that the entire carapace was "net stabilizing," which is evidenced by the fact that Bartol and colleagues never state this outright and only rarely measured net torque (and even then only in pitch, not in yaw).

→ **We take note of the reviewer's opinion. As the pitch torque graphs from the Bartol *et al.* papers show a net stabilisation by the carapace (see our previous response-to-review letter), in our opinion, their description of the hydrodynamic behaviour of the carapace as 'self-correcting' is a synonym of being 'net stabilising'.**

The authors have done a nice job revising how they address the Bartol et al experiments, but the paragraph where they summarize this (new lines 70-81) still sets up a framework of: (A) A hypothesis of self-correcting vortices was proposed by Bartol et al and (B) Van Wassenbergh et al rejected that hypothesis because they found the carapace to net-stabilizing. If a net-stabilizing carapace and self-correcting vortices are not mutually-exclusive hypotheses, I still don't think this framework is appropriate.

→ **The second reviewer has a similar comment. We now clarified this by replacing 'this hypothesis' (changes in orange) by: "However, the hypothesis concerning overall course stabilisation by the carapace was rejected by Van Wassenbergh et al. [11]."**

I leave it up to the authors and editors to decide if this point is important enough to warrant further revision.

I appreciate the opportunity to review this manuscript, and I congratulate the authors on a nice study that will advance the field.

Reviewer: 3

Comments to the Author(s)

Review comments on RSOS-200129

In this study, the authors investigated the yaw control of the boxfish by caudal fin. They demonstrated that the rigid body was unstable, while the caudal fin acted as both a stabilizer and rudder through an interplay with the rigid body. The data in this study is valuable, but I think the interpretation on the data ought to be improved.

1. The title only focuses on the 'stability' role of the caudal fin, but the manuscript actually tells that the caudal fin is both a stabilizer and rudder (stability and maneuverability). Hence, the title seems inaccurate and providing a lopsided first impression.

→ **We agree with this comment by the reviewer. Previous reviewers have opposed against the usage of "manoeuvrability" in the title because the analysis is limited to measuring torques in a static setting. We therefore suggest the following new title: "Modulating yaw with an unstable rigid body and a course-stabilising or steering caudal fin in the yellow boxfish (*Ostracion cubicus*)". We also changed some keywords to minimise overlap with words used in the title.**

2. My major concern on the authors' interpretation on their data is: what is stability/stabilizing? The authors seem considering this issue simply: when an including angle occurs between fish and oncoming flow (body angle α in the text), effects reducing this angle are stabilizing, while effects increasing this angle are destabilizing. However, we may imagine the following two circumstances: First, the boxfish is still and a gust flow comes laterally; Second, the boxfish is swimming forward while a gust flow comes laterally. In both circumstances, an angle occurs between fish and oncoming flow (for swimming condition, oncoming flow speed is the composite speed of swimming speed and gust flow speed). Neither increment nor decrement of this angle brings stability. Instead, stability means maintaining the current heading direction of fish, which corresponds to the ZERO torque points in Fig.7.

→ **We agree that in our definition of swimming stability, responses to lateral gusts may not be the most efficient for the tail poses we refer to as the ones causing the strongest stability, but this is an inherent trade-off: torques opposing yaw angles of attack will enable a straight swimming path in still water (which we define as high stability) but will**

inevitably cause stronger rotation towards sudden incoming flows from the side. Nevertheless, realignment with the swimming direction will occur once such (short) lateral gusts cease in our definition of stability, or similarly, when propulsors like the pectoral fins have finished generating some yaw impulse. To make it clearer that our definition of stability refers to the swimming trajectory of the fish, we changed 'stabilising' by 'course-stabilising' in the title. In addition, we changed this wording in the running text throughout the manuscript to clarify and mention at the end of the discussion that responses to lateral gust flows warrant further study.

Minor comments:

3. Line 70-81, in this paragraph, the authors actually list two hypotheses of the previous studies. First hypothesis: carapace induces stabilizing vortices; second hypothesis: carapace-induced stabilizing vortices may lead to overall stability. Rather than stating 'THIS hypothesis was rejected', the authors would better improve their description to clarify which hypothesis was rejected.
 - ➔ **We agree with the referee and adjusted the manuscript accordingly to highlight that the hypothesis regarding the overall stability by the carapace was rejected and not that the carapace induces stabilising vortices. We rewrote this section as follows (changes in orange; we also split the sentence to increase readability):**
 - Previous studies hypothesised that the flow over the carapace of boxfishes induces course-stabilising and self-corrective trimming vortices during pitching (i.e. rotation about the lateral axis) and yawing (i.e. rotation about the dorsoventral axis) [14,15,22,23]. This passive inherent course stabilisation by flow over the carapace (without fins) was hypothesised to dampen and counteract perturbations when swimming, and supposed to keep boxfishes on their swimming course whilst in turbulent waters, resulting in optimal stability [14,22,23,26]. However, **the hypothesis concerning overall course stabilisation by the carapace** was rejected by Van Wassenbergh *et al.* [11]. **They** showed that the body-induced vortices indeed exist, but that the overall impact on the pitch or yaw torque balance is relatively small: the body (i.e. carapace with a truncated part of the caudal peduncle) of both the yellow boxfish *Ostracion cubicus* (Linnaeus, 1758) and smooth trunkfish *Rhinesomus triquetter* (Linnaeus, 1758) generate destabilising hydrodynamic torques (i.e. are naturally hydrodynamically unstable).
4. Could the authors explain those large errors of the data points at 40 degrees in Fig.5(b)?
 - ➔ **The exact reason is not known to us, but this probably has to do with interference of vibrations between the different components of our measurement apparatus, or due to effects of unsteadiness of flows at certain angles. Nevertheless, in our view, the error bars are relatively small and do not affect the conclusions drawn from these data.**
5. Line 402, since the reference Farina & Summers [45] is not a research, but a news report based on the reference Van Wassenbergh [11], it should not appear in the discussion. If the authors think it's important, I suggest move it to somewhere in the introduction part, with highlight of the type of this reference.
 - ➔ **This 'news and views' reference contains original scientific views on this topic beyond reporting the 'news'. Consequently, similar to other review articles, we feel that it is an appropriate reference for a discussion section.**